# Adversarial Auto-Augment with Label Preservation: A Representation Learning Principle Guided Approach

**Kaiwen Yang**[1,4]    **Yanchao Sun**[2]    **Jiahao Su**[2]    **Fengxiang He**[4]
**Xinmei Tian**[1,3]    **Furong Huang**[2]    **Tianyi Zhou**[2]    **Dacheng Tao**[4,5]

University of Science and Technology of China[1]; University of Maryland, College Park[2]
Institute of Artificial Intelligence, Hefei Comprehensive National Science Center[3]
JD Explore Academy[4]; The University of Sydney[5]
kwyang@mail.ustc.edu.cn, xinmei@ustc.edu.cn,
{ycs, jiahaosu, furongh}@umd.edu
{fengxiang.f.he, tianyi.david.zhou, dacheng.tao}@gmail.com

## Abstract

Data augmentation is a critical contributing factor to the success of deep learning but heavily relies on prior domain knowledge which is not always available. Recent works on automatic data augmentation learn a policy to form a sequence of augmentation operations, which are still pre-defined and restricted to limited options. In this paper, we show that a prior-free autonomous data augmentation's objective can be derived from a representation learning principle that aims to preserve the minimum sufficient information of the labels. Given an example, the objective aims at creating a distant "hard positive example" as the augmentation, while still preserving the original label. We then propose a practical surrogate to the objective that can be optimized efficiently and integrated seamlessly into existing methods for a broad class of machine learning tasks, e.g., supervised, semi-supervised, and noisy-label learning. Unlike previous works, our method does not require training an extra generative model but instead leverages the intermediate layer representations of the end-task model for generating data augmentations. In experiments, we show that our method consistently brings non-trivial improvements to the three aforementioned learning tasks from both efficiency and final performance, either or not combined with strong pre-defined augmentations, e.g., on medical images when domain knowledge is unavailable and the existing augmentation techniques perform poorly. Code is available at: https://github.com/kai-wen-yang/LPA3.

## 1 Introduction

Data augmentation has emerged as an effective data pre-processing or data transformation step to mitigate overfitting [31], to encourage local smoothness [57], and to improve generalization [6] in machine learning pipelines such as deep neural networks. Notably, effective data augmentation, which incorporates class-related data invariance and enriches the in-class sample, is one of the key contributing factors for representation learning with weak or self supervision [9, 23].

Given a task, we aim to generate "good" augmentations efficiently. As part of the machine learning model pipeline, an *autonomous domain-agnostic* but *task-informed* data augmentation mechanism is desirable. However, a number of challenges exist. (**1**) Existing augmentation operators are usually hand-crafted based on domain expert knowledge, which is not always available in some domain [49]. For example, widely used augmentations for natural images are not effective on medical images. Moreover, the performance of those machine learning pipelines drastically varies with different choices of data augmentations. (**2**) Existing few autonomous augmentation approaches developed

36th Conference on Neural Information Processing Systems (NeurIPS 2022).

lately are neither fully autonomous nor universally applicable to varying domains. Although a few autonomous data augmentation approaches have been developed in recent years [14, 12], they train policies to produce a sequence of pre-defined augmentation operations and thus are not fully automated and are limited to a few domains. **(3)** Existing augmentations usually do not fully utilize the task feedback (i.e., task-agnostic) and may be sub-optimal for the targeted task. A class of automated data augmentation methods train an extra data generative model to generate new augmentations from scratch given a real-world example [2]. However, they require training a generative model, which is a non-trivial task in practice that may either relies on strong prior knowledge or a substantial increased number of training examples.

In this paper, we first investigate the conditions required to generate domain-agnostic but task-informed data augmentations. Consider a representation learning pipeline, we started from a probabilistic graphical model that describes the relations among the label $\mathbf{Y}$, the nuisance $\mathbf{N}$, the example $\mathbf{X}$, its augmentation $\mathbf{X}'$, and the latent representations $\mathbf{Z}$. We argue that a minimum-sufficient representation for the task preserves the label information but excludes other distractive information from the nuisance. We then investigate the conditions for an augmentation $\mathbf{X}'$ that results in learning such preferred representations. These conditions motivate an optimization objective that can be used to produce automated domain-agnostic but task-informed data augmentations for each example, without replying on pre-defined augmentation operators or specific domain knowledge. Consequently, our proposed optimization objective addresses all aforementioned challenges.

For practicality, we further propose a surrogate of the derived objective that can be efficiently computed from the intermediate-layer representations of the model-in-training. The surrogate is built upon the data likelihood estimation through perceptual distance [24] defined on the intermediate layers' representations. Specifically, our proposed surrogate objective maximizes the perceptual distance between $\mathbf{X}$ and $\mathbf{X}'$, under a label preserving constraint on the model prediction of $\mathbf{X}'$. This problem can be efficiently solved by optimizing its Lagaragian relaxation. Thereby, given $\mathbf{X}$ and its label $\mathbf{Y}$, the solution to our surrogate objective generates "hard positive examples" for $\mathbf{X}$ without loosing its label information. Once generated, $\mathbf{X}'$ is used to train the model towards producing the minimum-sufficient representation $\mathbf{Z}$ for the targeted task. Our proposed method, named *Label-Preserving Adversarial Auto-Augment (LP-A3)*, does not require any extra generative models such as Generative Adversarial Networks, unlike previous automated augmentation methods [40]. We further propose a sharpness-aware criterion selecting only the most informative examples to apply our auto-augmentation on so it does not cause expensive extra computation.

Our proposed LP-A3 is a general and autonomous data augmentation technique applicable to a variety of machine learning tasks, such as supervised, semi-supervised and noisy-label learning. Moreover, we demonstrate that it can be seamlessly integrated with existing algorithms for these tasks and consistently improve their performance. In experiments on the three learning tasks, we equip LP-A3 with existing methods and obtain significant improvement on both the learning efficiency and the final performance. The generated augmentations are optimized for the model-in-training in a target-task-aware manner and thus notably accelerate the slow convergence in computationally intensive tasks such as semi-supervised learning. It is worth noting that our augmentation can consistently bring improvement to tasks without domain knowledge or strong pre-defined augmentations such as medical image classification, on which previous image augmentations lead to performance degeneration.

## 2   Related work

**Hand-crafted vs. Autonomous Data Augmentations.** Most of the existing widely used data augmentations are hand-crafted based on domain expert knowledge [30, 39, 29, 9, 13]. For example, MoCo [16] and InstDis [47] create augmentations by applying a stochastic but pre-defined data augmentation function to the input. CMC [39] splits images across color channels. PIRL [29] generates data augmentations through random JigSaw Shuffling. CPC [30] renders strong data augmentations by utilizing RandAugment [12], which learns a policy producing a sequence of pre-defined augmentation operations selected from a pool [14]. AdvAA [60] designs a adversarial objective to learn the augmentation policy. [12, 14, 60] are all based on pre-defined operations which is not available in certain domains, and their objective cannot guarantee the label-preserving of the generated data which may lead to suboptimal performance. "InfoMin" principle of data augmentation is proposed [40] to minimize the mutual information between different views (equivalent to $\min I(\mathbf{X}, \mathbf{X}')$). However, their theory depends on access to a minimal sufficient encoder which may be difficult to obtain. In

contrast, we not only consider how to generate optimal views or augmentations, but also consider generating the minimal sufficient representation. The algorithm [40] deploys a generator to render augmentation (which may be costly to train especially on non-natural image domains), while we directly learn the augmentation through gradient descent w.r.t. the input.

**Information Theory for Representation Learning.** Information theory is introduced in deep learning to measure the quality of representations [42, 1]. The key idea is to use information bottleneck methods [41, 42] to encourage the learned representation being minimal sufficient. Mutual information objectives are commonly used in self-supervised learning. For example, InfoMax principle [27] used by many works aims to maximize the mutual information between the representation and the input [39, 4, 46]. But simply maximizing the mutual information does not always lead to a better representation in practice [43]. In contrast, InfoMin principle [40] minimizes the mutual information between different views. Both InfoMax and the InfoMin principles can be associated with our proposed representation learning criteria in Section 4, as they lead to sufficiency and minimality of the learned representation, respectively.

**Augmentation in Self-supervised Contrastive Learning.** Self-supervised Contrastive Representation Learning [30, 18, 47, 39, 35, 9] learn representation through optimization of a contrastive loss which pulls similar pairs of examples closer while pushing dissimilar example pairs apart. Creating multiple views of each example is crucial for the success of self-supervised contrastive learning. However, most of the data augmentation methods used in generating views, although sophisticated, are hand-crafted or not learning-based. Some use luminance and chrominance decomposition [39], while others use random augmentation from a pool of augmentation operators [47, 9, 4, 16, 53, 37, 62, 65]. Recently, adversarial perturbation based augmentation has been proposed to generate more challenging positives/negatives for contrastive learning [51, 19].

**Augmentation in Semi-supervised Learning.** Data augmentation plays an important role in semi-supervised learning, e.g., (1) consistency regularization [33, 34] enforces the model to produce similar outputs for a sample and its augmentations; (2) pseudo labeling [25] trains a model using confident predictions produced by itself [38] for unlabeled data. Data augmentations are critical [23, 7] because they determine both the output targets and input signals: (1) accurate pseudo labels are achieved by averaging the predictions over multiple augmentations; (2) weak augmentations (e.g., flip-and-shift) are important to produce confident pseudo labels, while strong augmentations [14, 12]) are used to train the model and expand the confidence regions (so more confident pseudo labels can be collected later). Data selection [55, 63] for high-quality pseudo labels is also critical and its criterion is estimated on augmentations, e.g., the confidence [8] or time-consistency [63] of each sample.

**Augmentation in Noisy-label Learning** Two primary challenges in noisy-label learning is clean label detection [28, 15, 20] and noisy label correction by pseudo labels [32, 3, 26]. Both significantly depend on the choices of data augmentations since the former usually relies on confidence thresholding and augmentations help rule out the overconfident samples, while the latter relies on the quality of semi-supervised learning. Moreover, as shown in previous works [26, 64], removing strong augmentations such as RandAugment can considerably degenerate the noisy label learning performance.

## 3   Preliminaries

**Basics of Information Theory** Our analyses make frequent use of information theoretical quantities [11]. Given a joint distribution $P_{\mathbf{X},\mathbf{Y}}$ and its marginal distributions $P_{\mathbf{X}}$, $P_{\mathbf{Y}}$, we define their *entropy* as $H(\mathbf{X},\mathbf{Y}) = \mathrm{E}_{\mathbf{X},\mathbf{Y}}[-\log P(x,y)]$, $H(\mathbf{X}) = \mathrm{E}_{\mathbf{X}}[-\log P(x)]$, and $H(\mathbf{Y}) = \mathrm{E}_{\mathbf{Y}}[-\log P(y)]$. Furthermore, we define the conditional entropy of $\mathbf{X}$ given $\mathbf{Y}$ as $H(\mathbf{X}|\mathbf{Y}) = \mathrm{E}_{\mathbf{Y}}[-\log H(\mathbf{X}|y)] = H(\mathbf{X},\mathbf{Y}) - H(\mathbf{Y})$. Finally, we define the mutual information between $\mathbf{X}$ and $\mathbf{Y}$ as $I(\mathbf{X} \wedge \mathbf{Y}) = H(\mathbf{X}) - H(\mathbf{X}|\mathbf{Y}) = H(\mathbf{X}) + H(\mathbf{Y}) - H(\mathbf{X},\mathbf{Y})$.

**Notations and Problem Setup.** In this paper, we use bold capital letters (e.g., $\mathbf{X}, \mathbf{Y}$) to denote random variables, lowercase letters (e.g., $x, y$) to denote their realizations, and curly capital letters (e.g., $\mathcal{X}, \mathcal{Y}$) to denote the corresponding sample spaces.

Since we mainly consider supervised and semi-supervised problems, we define Let $P_{\mathbf{X},\mathbf{Y}}$ be the joint distribution of data *observation* $\mathbf{X}$ and *label* $\mathbf{Y}$, where $\mathbf{X}$ is a random vector taking values on a finite observation space $\mathcal{X}$ (e.g., images) and $\mathbf{Y}$ is a discrete random variable taking values on the label space $\mathcal{Y}$ (e.g., classes). Our goal is to learn a classifier to predict $y \in \mathcal{Y}$ from an observation $x \in \mathcal{X}$.

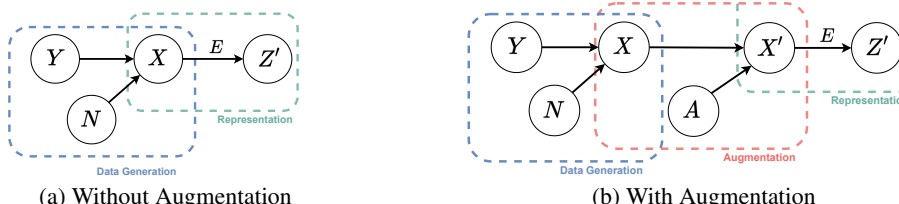

|                                                |                                              |
|:----------------------------------------------:|:--------------------------------------------:|
| (a) Without Augmentation                       | (b) With Augmentation                        |

Figure 1: **Probabilistic graphical models** of representation learning.

**Task-nuisance Decomposition.** To advance the analysis, we decouple the randomness in $\mathbf{X}$ into two parts, one pertaining to the label and another independent to the label. Concretely, we define a random variable *nuisance* $\mathbf{N}$ such that **1)** the nuisance $\mathbf{N}$ is independent to the label $\mathbf{Y}$, i.e., $\mathbf{N} \perp\!\!\!\perp \mathbf{Y}$; and **2)** the observation $\mathbf{X}$ is a deterministic function of the nuisance $\mathbf{N}$ and the label $\mathbf{Y}$, i.e., $\mathbf{X} = g(\mathbf{Y}, \mathbf{N})$ for some $g$. Lemma 3.1 demonstrates that such a random variable always exists.

**Lemma 3.1** (Task-nuisance Decomposition [1]). *Given a joint distribution $P_{\mathbf{X}, \mathbf{Y}}$, where $\mathbf{Y}$ is a discrete random variable, we can always find a random variable $\mathbf{N}$ independent of $\mathbf{Y}$ such that $\mathbf{X} = d(\mathbf{Y}, \mathbf{N})$, for some deterministic function $d$.*

*Remarks.* We can rewrite the conditions of task-nuisance decomposition in terms of information theory. **1)** Since the nuisance $\mathbf{N}$ is independent to the label $\mathbf{Y}$, we have $I(\mathbf{Y} \wedge \mathbf{N}) = 0$; and **2)** Since the nuisance $\mathbf{N}$ and the label $\mathbf{Y}$ determines the observation $\mathbf{X}$, we have $H(\mathbf{X}|\mathbf{Y}, \mathbf{N}) = 0$.

## 4 Principles of Representation Learning: Theoretical Interpretation

### 4.1 What Is A Good Representation?

In real-world applications, the observation $\mathbf{X}$ is usually complex in a high-dimensional space $\mathcal{X}$, making it hard to directly learn a good classifier for $\mathbf{Y}$. To remedy this curse of dimensionality, it is important to learn a good representation of $\mathbf{X}$, i.e., learn an encoder $E(\cdot)$ that maps the high-dimensional observation $\mathbf{X}$ into a low-dimensional representation $\mathbf{Z}$. We illustrate the process of data generation and representation learning by a probabilistic graphical model as shown in Figure 1a.

An ideal encoder should keep the important information from $\mathbf{X}$ (e.g. label-relevant information) and maximally discard the noise or nuisance of $\mathbf{X}$, such that it is much easier to learn a classifier from $\mathbf{Z}$ than from $\mathbf{X}$. Based on the above intuition, we define an $\epsilon$-optimal representation of $\mathbf{X}$, which has sufficient information for classifying w.r.t. $\mathbf{Y}$, while remaining little information of the nuisance.

**Definition 4.0.1** ($\epsilon$-Minimal Sufficient Representation ($\epsilon$-Optimal Representation)). *For a Markov chain $\mathbf{Y} \rightarrow \mathbf{X} \rightarrow \mathbf{Z}$, we say thata representation $\mathbf{Z}$ of $\mathbf{X}$ is sufficient for $\mathbf{Y}$ if $I(\mathbf{Z} \wedge \mathbf{Y}) = I(\mathbf{X} \wedge \mathbf{Y})$, and $\mathbf{Z}$ is $\epsilon$-minimal sufficient for $\mathbf{Y}$ if $\mathbf{Z}$ is sufficient and $I(\mathbf{Z} \wedge \mathbf{X}) \leq I(\tilde{\mathbf{Z}} \wedge \mathbf{X}) + \epsilon$ for all $\tilde{\mathbf{Z}}$ satisfying $I(\tilde{\mathbf{Z}} \wedge \mathbf{Y}) = I(\mathbf{X} \wedge \mathbf{Y})$.*

*Remark.* Due to the property of mutual information, we have $0 \leq \epsilon \leq H(\mathbf{X})$. The lower $\epsilon$ is, the more "minimal" the representation is. When $\epsilon = 0$, the representation is minimal sufficient, which is a desirable property as characterized by many prior works [42, 1].

Definition 4.0.1 characterizes how good a sufficient representation is, based on how much redundant information is remained. Recall that $\mathbf{X}$ comes from a deterministic function of label $\mathbf{Y}$ and nuisance $\mathbf{N}$. The redundancy of $\mathbf{Z}$ can also be measured by the mutual information between $\mathbf{Z}$ and $\mathbf{N}$. Achille et al. [1] prove that if a representation $\mathbf{Z}$ is sufficient and is invariant to nuisance $\mathbf{N}$, i.e., $I(\mathbf{Z} \wedge \mathbf{N}) = 0$, then $\mathbf{Z}$ is also minimal. However, since $\mathbf{N}$ is not known, it is hard to directly encourage the representation to be invariant to $\mathbf{N}$.

Can we learn an $\epsilon$-minimal sufficient representation in a principled way? Inspired by the recent success of data augmentation techniques in self-supervised learning and semi-supervised learning, we find that data augmentation can implicitly encourage the representation to be invariant to the nuisance $\mathbf{N}$. However, most augmentation methods are driven by pre-defined transformations, which do not necessarily render a minimal sufficient representation. In the next section, we will analyze the effects of data augmentation in representation learning in details.

## 4.2 Proper Data Augmentation Leads to (Near-)Optimal Representation

In this section, we investigate the role of data augmentation for learning good representations. We first make the following mild assumption on the underlying relationship between $\mathbf{X}$ and $\mathbf{Y}$.

**Assumption 4.1.** *There exists a deterministic function $\pi : \mathcal{X} \to \mathcal{Y}$, i.e., $H(\mathbf{Y}|\mathbf{X}) = 0$.*

Assumption 4.1 requires that there exists a "perfect classifier" that identifies the label $y$ of the observation $x$ with no error, which is common in practice. Note that for data with ambiguity, a tie breaker can be used to map each observation to a unique label. Therefore, Assumption 4.1 is realistic.

Let $g$ be a deterministic augmentation function such that $\mathbf{X}' := g(\mathbf{X}, \mathbf{A})$ is the augmented data, where $\mathbf{A}$ is a random variable denoting the augmentation selection. For example, if $\mathbf{X} = x$ is an image sample, $\mathbf{A} = a$ is the augmentation "rotate by 90 degree", then $\mathbf{X}' = x'$ is the corresponding rotated image sample. We learn an encoder $E(\cdot)$ that maps the augmented data $\mathbf{X}'$ to a representation $\mathbf{Z}'$. With this augmentation processes, the graphical model in Figure 1a is updated to Figure 1b.

We show in the theorem below that if the augmentation process preserves the information of $\mathbf{Y}$, $\mathbf{Z}'$ can be sufficient for $\mathbf{Y}$. Furthermore, if the augmented data $\mathbf{X}'$ contains no information of the original nuisance $\mathbf{N}$, $\mathbf{Z}'$ will be invariant to $\mathbf{N}$ and thus will become a minimal sufficient representation.

**Theorem 4.2.** *Consider label variable $\mathbf{Y}$, observation variable $\mathbf{X}$ and nuisance variable $\mathbf{N}$ satisfying Assumption 4.1. Let $\mathbf{A}$ be the augmentation variable, $\mathbf{X}'$ be the augmented data, and $\mathbf{Z}^*$ be the solution to*

$$
\begin{aligned}
\mathrm{argmax}_{\mathbf{Z}'} \quad & I(\mathbf{Z}' \wedge \mathbf{X}') \text{ or } I(\mathbf{Z}' \wedge \mathbf{Y}) \\
\text{subject to} \quad & I(\mathbf{Z}' \wedge \mathbf{A}) = 0.
\end{aligned}
\tag{1}
$$

*Then, $\mathbf{Z}^*$ is a $\epsilon$-minimal sufficient representation of $\mathbf{X}$ for label $\mathbf{Y}$ if the following conditions hold:*
***Condition (a):*** *$I(\mathbf{X}' \wedge \mathbf{Y}) = I(\mathbf{X} \wedge \mathbf{Y})$ ($\mathbf{X}'$ is an in-class augmentation) and*
***Condition (b):*** *$I(\mathbf{X}' \wedge \mathbf{N}) \leq \epsilon$ ($\mathbf{X}'$ does not remain much information about $\mathbf{N}$).*

**Remarks.** (1) The objective of learning $\mathbf{Z}^*$ can be either task-independent (maximizing $I(\mathbf{Z}' \wedge \mathbf{X}')$), or task-dependent (maximizing $I(\mathbf{Z}' \wedge \mathbf{Y})$). The former matches the "InfoMax" principle commonly used in self-supervised learning works [27, 18], while the latter can be achieved by supervised training (e.g., learning a classifier of $\mathbf{Z}$ for $\mathbf{Y}$ with cross-entropy loss).
(2) When Condition (b) holds for $\epsilon = 0$, representation $\mathbf{Z}'$ is optimal (minimal sufficient).

Theorem 4.2, proved in Appendix B.1, shows that if we have a good augmentation that maximally perturbs the label-irrelevant information while keeps the label-relevant information, then the representation learned on the augmented data can be minimal sufficient. Theorem 4.2 serves as a principle of constructing augmentation. Based on this principle, we propose an auto-augment algorithm in Section 5, and verify the algorithm in a wide range tasks in Section 6.

## 5 Proposed Methods

In this section, we introduce our data augmentation and how to obtain the augmentation using the representation learning network $F(\cdot)$. Then we show how to plug our augmentation into the representation learning procedure of $F(\cdot)$.

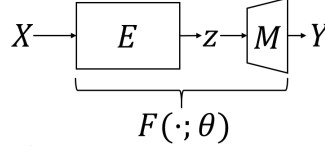

Figure 2: Network architecture.

### 5.1 Label-Preserving Adversarial Auto-Augment (**LP-A3**)

As illustrated in the previous section, an ideal data augmentation $\mathbf{X}'$ for representation learning should contain as little information about nuisance $\mathbf{N}$ as possible while still keeping all the information about class $\mathbf{Y}$. Since $\mathbf{N}$ is not observed, we transfer the objective $\min_{\mathbf{X}'} I(\mathbf{X}' \wedge \mathbf{N})$ into $\min_{\mathbf{X}'} I(\mathbf{X}' \wedge \mathbf{X})$ since $I(\mathbf{X}' \wedge \mathbf{X}) = I(\mathbf{X}' \wedge \mathbf{N}) + I(\mathbf{X}' \wedge \mathbf{Y})$ and $I(\mathbf{X}' \wedge \mathbf{Y})$ is a constant under the constraint $I(\mathbf{X}' \wedge \mathbf{Y}) = I(\mathbf{X} \wedge \mathbf{Y})$. Thus the optimization problem is:

$$
\min_{\mathbf{X}'} I(\mathbf{X}' \wedge \mathbf{X}) \quad \text{s.t. } I(\mathbf{X}' \wedge \mathbf{Y}) = I(\mathbf{X} \wedge \mathbf{Y}).
\tag{2}
$$

**Implementation of Mutual Information.** To solve Equation (2), computing the mutual information terms $I(\mathbf{X}' \wedge \mathbf{X})$, $I(\mathbf{X}' \wedge \mathbf{Y})$ and $I(\mathbf{X} \wedge \mathbf{Y})$ is required. Next, we will show how to compute these terms using a neural net classifier $F(\cdot; \theta)$, parameterized by $\theta$, that consists of two components:

a representation encoder $E(\cdot)$ and a predictor $M(\cdot)$. Specifically, $F(\cdot; \theta) = M(E(\cdot))$, where the representation encoder $E(\cdot)$ maps input $\mathbf{X}$ into representation $\mathbf{Z}$, and the predictor $M(\cdot)$ predicts the class of $\mathbf{Z}$. This is demonstrated in Figure 2.

*Constraint implementation.* Since $I(\mathbf{X}' \wedge \mathbf{Y}) = H(\mathbf{Y}) - H(\mathbf{Y}|\mathbf{X}')$ and $I(\mathbf{X} \wedge \mathbf{Y}) = H(\mathbf{Y}) - H(\mathbf{Y}|\mathbf{X})$, we can remove the $H(\mathbf{Y})$ term in both sides and turn the constraint into $H(\mathbf{Y}|\mathbf{X}) = H(\mathbf{Y}|\mathbf{X}')$. Thus we only need to compute the conditional entropy of $\mathbf{Y}$ given $\mathbf{X}$ or $\mathbf{X}'$, which can be approximated through the neural net classifier: $H(\mathbf{Y}|\mathbf{X}) = \mathrm{E}_{\mathbf{X},\mathbf{Y}}\left[-\log P(y|x)\right] \approx \mathrm{E}_{\mathbf{X},\mathbf{Y}}\left[-\log(F(x;\theta)[y])\right]$, where we use softmax class probability $F(x;\theta)[y]$ to approximate the likelihood $P(y|x)$. And $H(\mathbf{Y}|\mathbf{X}')$ can be computed similarly.

*Objective implementation.* Then we show how to compute the objective $I(\mathbf{X}' \wedge \mathbf{X})$. Since $I(\mathbf{X}' \wedge \mathbf{X}) = H(\mathbf{X}) - H(\mathbf{X}|\mathbf{X}')$ where $H(\mathbf{X})$ is not related to $\mathbf{X}'$ and thus can be neglected, we only need to compute $H(\mathbf{X}|\mathbf{X}') = \mathrm{E}_{\mathbf{X},\mathbf{X}'}\left[-\log P(x|x')\right]$. We use the Learned Perceptual Image Patch Similarity (LPIPS) [59] between $x$ and $x'$ to compute the data likelihood $P(x|x')$ since LPIPS distance is a widely used metric to measure the data similarity in data generative model field [21, 58] and many previous work has shown that LPIPS distance is the best surrogate for human comparisons of similarity [59, 24], compared with any other distance including $\ell_2$ and $\ell_\infty$ distance. Although such surrogate may have error, it worth noting that Theorem 4.2 allows the surrogate to have $\epsilon$ error. The LPIPS distance is defined by the $\ell_2$ distance of stacked feature maps from a neural network. Here we use $F(\cdot; \theta)$ to compute the LPIPS distance. Let $F(\cdot; \theta)$ has $L$ layers and $\widehat{F}_l(\cdot; \theta)$ denotes these channel-normalized activations at the $l$-th layer of the network. Next, the activations are normalized again by layer size and flattened into a single vector $\phi(x) \triangleq (\frac{\widehat{F}_1(x;\theta)}{\sqrt{w_1 h_1}}, ..., \frac{\widehat{F}_L(x;\theta)}{\sqrt{w_L h_L}})$, where $w_l$ and $h_l$ are the width and height of the activations in layer $l$, respectively. The LPIPS distance between input $x$ and the augmentation $x'$ is then defined as:

$$\mathrm{LPIPS}(x, x') \triangleq \|\phi(x) - \phi(x')\|_2. \tag{3}$$

**Constraint Relaxation for Efficiency.** Now, given an input $x$, its data augmentation $x'$ can be computed by solving the following optimization problem using the neural network $F(\cdot; \theta)$ in practice:

$$\min_{x'} -\|\phi(x) - \phi(x')\|_2 \quad \text{s.t. } \log F(x';\theta)[y] = \log F(x;\theta)[y]. \tag{4}$$

The equality constraint in Equation (4) is too strict to solve since it may be inefficient to search for an $x'$ that exactly satisfies $\log F(x';\theta)[y] = \log F(x;\theta)[y]$. Thus we relax the constraint with a small $\sigma$ and change the constaint into: $\log F(x';\theta)[y] \geq \log F(x;\theta)[y] - \sigma$. It's worth noting that if $\sigma$ is sufficiently small, the label is still well preserved. There is a trade-off to the value of $\sigma$, we search $\sigma$ to find a sweet spot where the problem is practical to solve and meanwhile the label is well preserved.

There are many off-the-shelf methods that solve Equation (4), and here we apply the Fast Lagrangian Attack Method [24] as a demonstration. We initialize $x'$ by $x$ plus a uniform noise. And we find the optimal $x'$ by solving the following the Lagrangian multiplier function and gradually scheduling the value of the multiplier $\lambda$:

$$\min_{x'} -\|\phi(x) - \phi(x')\|_2 + \lambda \max(0, \log F(x;\theta)[y] - \log F(x';\theta)[y] - \sigma) \tag{5}$$

The detailed procedure of the algorithm can be found in Appendix 2. The algorithm has a similar form as adversarial attack [61, 52] in that they both find an optimal augmentation $x'$ by adding perturbations to the original image $x$. However, the difference is that we aim to generate hard augmentation that preserves the label, while adversarial attack aims to change the class label.

## 5.2 Plugging LP-A3 into a Representation Learning Task

One primary advantage of LP-A3 is that it only requires a neural net $F(\cdot; \theta)$ to produce the augmentation and $F(\cdot; \theta)$ can be the current representation learning model, so we can plug LP-A3 into any representation learning procedure requiring no additional parameters, which is plug-and-play and parameter-free. At each step, we first fix $F(\cdot; \theta)$ to generate the augmentation $x'$ by solving Equation (5) using Algorithm 2. And then we train $F(\cdot; \theta)$ by running the original representation learning algorithm using our augmentation $x'$.

**Data selection.** It is not necessary to find hard positives for every sample. To save more computation, we can apply a sharpness-aware criterion, i.e., time-consistency (TCS) [63], to select the most

informative data ($\tau\%$ data with the lowest TCS in Algorithm 1) that have sharp loss landscapes, which indicate the existence of nearby hard positives, and we only apply LP-A3 to them. It reduces the computational cost without degrading the performance because (1) the improvement brought by augmentations is limited for examples whose loss already reaches a flat minimum, while the model does not generalize well near examples with a sharp loss landscapes; and (2) the hard positives for examples with flat loss landscape are distant from the original ones and might introduce extra bias to the training.

LP-A3 is compatible with any representation learning task minimizing a loss $L : \mathcal{X} \times \mathcal{Y} \times \mathcal{W} \to \mathcal{R}_+$, which takes in a data batch and a model to output a loss value. $\mathcal{Y}$ here denotes the groundtruth label for labeled data and pseudo label for unlabeled data. The pseudo-code of plugging LP-A3 into the representation learning procedure with TCS-based data selection is provided in Algorithm 1.

---

**Algorithm 1** Plug LP-A3 into any representation learning procedure

---

**Input:** Loss for the targeted task $L : \mathcal{X} \times \mathcal{Y} \times \mathcal{W} \to \mathcal{R}_+$; training data $(\mathcal{X}, \mathcal{Y})$; neural network $F(\cdot; \theta)$; class preserving margin $\epsilon$; data selection ratio $\tau$; learning rate $\eta$;
**Output:** Model parameter $\theta$ trained with LP-A3
1: **while** *not converged* **do**
2:     Sample batch $\mathcal{B} = \{(x_1, y_1), ..., (x_b, y_b)\} \sim (\mathcal{X}, \mathcal{Y})$;
3:     Data selection: $\mathcal{S} \leftarrow \tau\%$ data with the lowest TCS in $\mathcal{B}$;
4:     LP-A3: Freeze $\theta$ and solve Equation (5) using Algorithm 2 for every sample in $\mathcal{S}$, resulting in an augmented set $\mathcal{A} = \{(x'_1, y_1), ..., (x'_m, y_m)\}$ of size $m = |\mathcal{S}|$;
5:     Learning with LP-A3 augmented data and original data: $\theta \leftarrow \theta - \eta[\nabla_\theta L(\mathcal{B}; \theta) + \nabla_\theta L(\mathcal{A}; \theta)]$;
6: **end while**

---

# 6 Experiments

In this section, we apply LP-A3 as a data augmentation method to several popular methods for three different learning tasks, i.e., (1) semi-supervised classification; (2) noisy-label learning and (3) medical image classification. In all the experiments, LP-A3 can (1) consistently improve the convergence and test accuracy of existing methods and (2) autonomously produce augmentations that bring non-trivial improvement even without any domain knowledge available. A walk-clock time comparison is given in the Appendix, showing LP-A3 effectively reduces the computational cost. In addition, we conduct a thorough sensitivity study of LP-A3 by changing (1) label-preserving margin and (2) data selection ratio on the three tasks. More experimental details can be found in the Appendix.

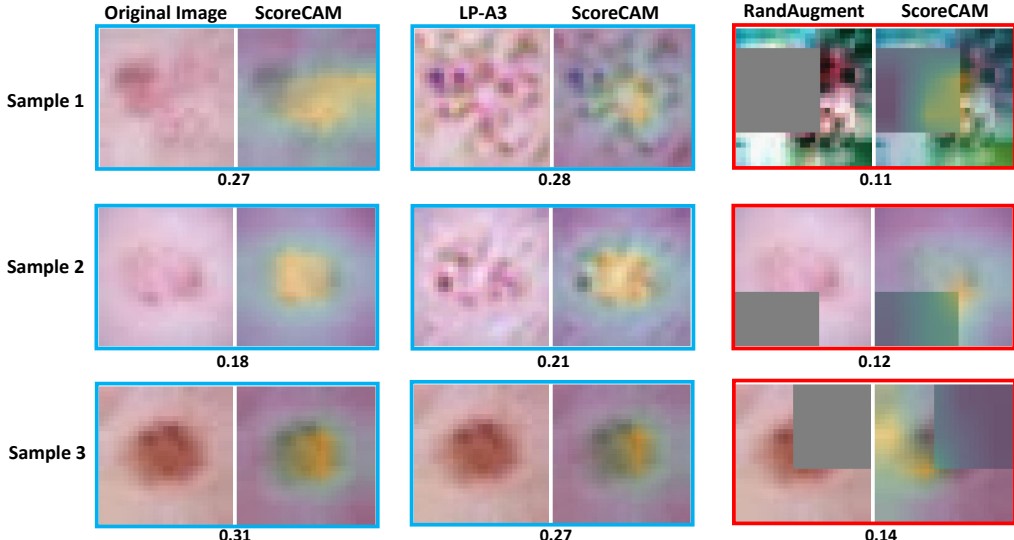

Figure 3: **Visualization of medical image augmentations** on the test set of DermaMnist. Blue (red) bounding box marks the correct (wrong) prediction of a ResNet18 classifier and its confidence on the groundtruth class is reported beneath the box. ScoreCAM [44] produces a heatmap highlighting important areas (by yellow color) of an image that a neural net mainly relies on to make the prediction.

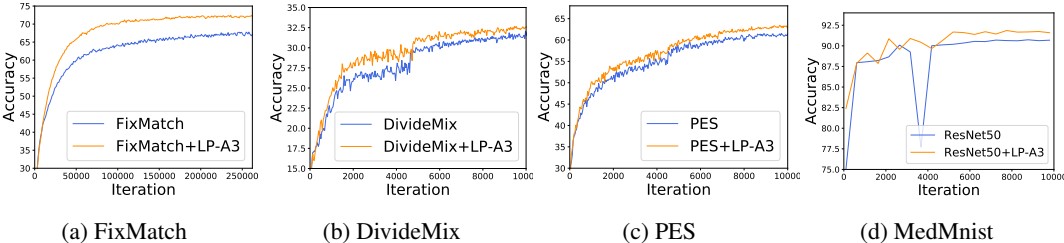

| (a) FixMatch | (b) DivideMix | (c) PES | (d) MedMnist |

Figure 4: **Convergence Curve** when applying LP-A3 to different tasks and baselines.

## 6.1 Medical Image Augmentations produced by LP-A3 vs. RandAugment

We visualize data augmentations generated by LP-A3 and RandAugment [12] on the testset of DermaMnist [50] with a ResNet-18 classifier and its confidence on the groundtruth class in Fig. 3. We also use ScoreCAM [44] as an interpretation method to highlight the area in each image that the classifier relies on to make the prediction. we find that LP-A3 preserves relevant derma areas highlighted by ScoreCAM and they are consistent with those in the original image. On the contrary, RandAugment changes the color or occludes those derma areas, resulting in highly different ScoreCAM heatmaps and hence wrong predictions (red bounding box in Fig. 3). Instead, LP-A3 can preserve the class information and mainly perturb the class-unrelated area in the original image.

## 6.2 Applying LP-A3 to Three Different Representation Learning Tasks

Here we apply LP-A3 to three different tasks by pluging LP-A3 to existing baselines of each task. Fig. 4 shows that LP-A3 greatly speeds up the convergence of each baseline.

**Semi-supervised learning** To evaluate how LP-A3 improves the learning without sufficient labeled data, we conduct experiments on semi-supervised classification on standard benchmarks including CIFAR [22] and STL-10 [10] where only a very small amount of labels are revealed. We apply LP-A3 in FixMatch [36] and compare it with the original FixMatch and InfoMin [40], a learnable augmentation method for semi-supervised learning. Their results are reported in Table 1, where LP-A3 consistently improves FixMatch and the improvement becomes more significant if reducing the labeled data. It's worth noting that the original FixMatch already employs a carefully designed set of pre-defined augmentations [13] that have been tuned to achieve the best performance, indicating that LP-A3 is complementary to existing data augmentations. Moreover, LP-A3 also outperforms InfoMin by a large margin ($> 5\%$), which indicates that LP-A3 is also superior to existing learnable augmentations.

Table 1: **Semi-supervised Learning** performance on CIFAR with different amounts of labeled data. [§] denotes results reproduced using the official code. FixMatch and LP-A3 are trained for $2^{18}$ SGD steps. InfoMin's results on CIFAR are missing since their paper only reports the result on STL-10. Error bars (mean and std) are computed over three random trails.

| Dataset | CIFAR10 | | | CIFAR100 | | | STL-10 |
|---|---|---|---|---|---|---|---|
| # Label | 40 | 250 | 4000 | 400 | 2500 | 10000 | 1000 |
| InfoMin (RGB) [40] | - | - | - | - | - | - | 86.0 |
| InfoMin (YDbDr) [40] | - | - | - | - | - | - | 87.0 |
| FixMatch [36][§] | 89.51±3.14 | 93.81±0.29 | 94.66±0.13 | 49.30±2.45 | 67.21±0.94 | 74.31±0.35 | 91.59±0.16 |
| FixMatch [36] + LP-A3 | **92.39±1.21** | **94.03±0.31** | **95.11±0.17** | **56.16±1.82** | **72.23±0.57** | **77.11±0.16** | **92.63±0.14** |

**Noisy-label Learning** Data augmentation is critical to noisy-label learning by providing different views of data to prevent neural nets from overfitting to noisy labels. We apply LP-A3 to two state-of-the-art methods DivideMix [26] and PES [5] on CIFAR with different ratios of noise labels. LP-A3 can consistently improve the performance of these two SoTA methods and the improvement is more significant in more challenging cases with higher noise ratios, e.g., on CIFAR100 with 90% of labels to be noisy, LP-A3 improves PES by $\geq 15\%$ (Table 2).

**Medical Image Classification** To evaluate the performance in specific areas without domain knowledge, we compare LP-A3 with existing data augmentations on medical image classification tasks from MedMNIST [50], which is composed of several sub-dataset with various styles of medical images. We compare our LP-A3 with RandAugment [13] on training ResNet-18 and ResNet-50 [17]. We report the results in Table 3, where RandAugment designed for natural images fails to improve

Table 2: **Noisy-label learning** performance on CIFAR with different ratios of symmetric label noises. § denotes the results reproduced by the official code. Error bars (mean and std) are computed over three random trails.

| Dataset | CIFAR10 | | | CIFAR100 | | |
|---|---|---|---|---|---|---|
| Noise Ratio | 50% | 80% | 90% | 50% | 80% | 90% |
| Mixup [56] | 87.1 | 71.6 | 52.2 | 57.3 | 30.8 | 14.6 |
| P-correction [54] | 88.7 | 76.5 | 58.2 | 56.4 | 20.7 | 8.8 |
| M-correlation [3] | 88.8 | 76.1 | 58.3 | 58.0 | 40.1 | 14.3 |
| DivideMix [26] | 94.4 | 92.9 | 75.4 | 74.2 | 59.6 | 31.0 |
| DivideMix+LP-A3 | 94.89±0.05 | **93.70±0.19** | 79.35±1.33 | 74.12±0.23 | 61.00±0.34 | 32.55±0.25 |
| PES§ [5] | 94.89±0.12 | 92.15±0.23 | 84.98±0.36 | 74.19±0.23 | 61.47±0.38 | 21.15±3.15 |
| PES+LP-A3 | **95.10±0.14** | 93.26±0.21 | **87.71±0.36** | **74.57±0.25** | **62.98±0.49** | **40.61±1.10** |

Table 3: **Medical Image Classification** on MedMNIST [50]. All the models are trained for 100 epochs. Error bars (mean and std) are computed over three random trails.

| Method | PathMNIST | DermaMNIST | TissueMNIST | BloodMNIST |
|---|---|---|---|---|
| ResNet-18 | 94.34±0.18 | 76.14±0.09 | 68.28±0.17 | 96.81±0.19 |
| ResNet-18+RandAugment | 93.52±0.09 | 73.71±0.33 | 62.03±0.14 | 95.00±0.21 |
| ResNet-18+LP-A3 | **94.42±0.24** | **76.22±0.27** | **68.63±0.14** | **96.97±0.06** |
| ResNet-50 | 94.47±0.38 | 75.24±0.27 | 69.69±0.23 | 96.91±0.06 |
| ResNet-50+RandAugment | 94.02±0.37 | 71.65±0.30 | 65.13±0.33 | 95.14±0.06 |
| ResNet-50+LP-A3 | **94.57±0.07** | **75.71±0.22** | **69.89±0.08** | **97.01±0.32** |
| | OctMNIST | OrganAMNIST | OrganCMNIST | OrganSMNIST |
| ResNet-18 | 78.67±0.26 | 94.21±0.09 | 91.81±0.12 | 81.57±0.07 |
| ResNet-18+RandAugment | 76.00±0.24 | 94.18±0.20 | 91.38±0.14 | 80.52±0.32 |
| ResNet-18+LP-A3 | **80.27±0.54** | **94.73±0.21** | **92.41±0.22** | **82.28±0.38** |
| ResNet-50 | 78.37±0.52 | 94.31±0.14 | 91.80±0.14 | 81.11±0.21 |
| ResNet-50+RandAugment | 76.63±0.58 | 94.59±0.17 | 91.10±0.12 | 80.47±0.37 |
| ResNet-50+LP-A3 | **79.40±0.36** | **94.95±0.19** | **92.16±0.23** | **82.15±0.08** |

the performance in this scenario. In contrast, LP-A3 does not rely on any domain knowledge brings improvement to all the datasets, especially for OctMNIST where the improvement is over $1\%$. The results indicate that hand-crafted strong data augmentations do not generalize to all domains but LP-A3 can autonomously produce augmentations guided by our representation learning principle without relying on any domain knowledge.

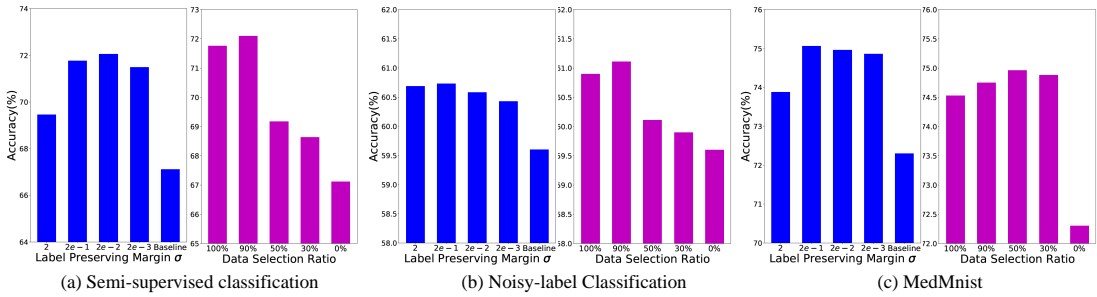

(a) Semi-supervised classification     (b) Noisy-label Classification     (c) MedMnist

Figure 5: **Sensitivity Analysis** of label preserving margin $\sigma$ and data selection ratio.

## 6.3 Sensitivity Analysis of Hyperparameters

**Label preserving margin $\sigma$:** We evaluate how LP-A3 performs with different label preserving margin $\sigma$ on the three tasks. The results are presented in Fig. 5, where a reverse U-shape is observed. And LP-A3 using all the evaluated $\sigma$ outperforms baselines, which indicates LP-A3 is robust to $\sigma$.

**Data selection ratio:** We evaluate the performance of LP-A3 with different amount of data selected on the three tasks. As shown in Fig. 5, selecting all the data does not perform the best since some data' augmentations are useless to apply data augmentation. Moreover, selecting only $30\%$ data to apply LP-A3 can outperform all baselines by a large margin, especially on MedMNIST where the improvement is $\geq 2\%$, which verifies the effectiveness of LP-A3 and our data selection method.

# 7 Conclusion

In this paper, we study how to automatically generate domain-agnostic but task-informed data augmentations. We first investigate the conditions required for augmentations leading to representations that preserves the task (label) information and then derive an optimization objective for the augmentations. For practicality, we further propose a surrogate of the derived objective that can be efficiently computed from the intermediate-layer representations of the model-in-training. The surrogate is built upon the data likelihood estimation through perceptual distance. This leads to LP-A3, a general and autonomous data augmentation technique applicable to a variety of machine learning tasks, such as supervised, semi-supervised and noisy-label learning. In experiments, we demonstrate that LP-A3 can consistently bring improvement to SoTA methods for different tasks even without domain knowledge. In future work, we will extend LP-A3 to more learning tasks and further improve its efficiency.

# Acknowledgements

This work was supported by, the Major Science and Technology Innovation 2030 "New Generation Artificial Intelligence" key project under Grant 2021ZD0111700, NSFC No. 61872329, No. 62222117, and the Fundamental Research Funds for the Central Universities under contract WK3490000005. Huang, Sun and Su are supported by NSF-IIS-FAI program, DOD-ONR-Office of Naval Research, DOD-DARPA-Defense Advanced Research Projects Agency Guaranteeing AI Robustness against Deception (GARD). Huang is also supported by Adobe, Capital One and JP Morgan faculty fellowships. We acknowledge the support of GPU cluster built by MCC Lab of Information Science and Technology Institution, USTC.

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
