# Supplementary Material

## A  Algorithmic Details

### A.1  Data Selection Via Time-Consistency

We use time-consistency (TCS) [63] to select informative sample to apply our augmentation, which computes the consistency of the output distribution for each sample along the training procedure. Specifically, TCS metric $c^t(x)$ for an individual sample is an negative exponential moving average of $a^t(x)$ over training history before $t$:

$$c^t(x) = \gamma_c\left(-a^t(x)\right) + (1 - \gamma_c)\, c^{t-1}(x) \tag{6}$$

$$a^t(x) \triangleq D_{KL}\left(F^{t-1}(x)\|F^t(x)\right) + \left|\log\frac{F^{t-1}(x)[y^{t-1}(x)]}{F^t(x)[y^{t-1}(x)]}\right| \tag{7}$$

where $D_{KL}(\cdot\|\cdot)$ is Kullback–Leibler divergence, $y^{t-1}(x)$ is pesudo label (for unlabeled data) or real label (for labeled data) for $x$ at step $t$ and $\gamma_c \in [0, 1]$ is a discount factor. Intuitively, the KL-divergence between output distributions measures how consistent the output is between two consecutive steps, and a moving average of $a^t(x)$ naturally captures inconsistency of $x$ over time quantify. And larger $c^t(x)$ means better time-consistency. We select top $\tau\%$ sample with the lowest TCS to apply our data augmentation because samples with small TCS tend to have sharp loss landscapes. These samples provide more informative gradients than others and applying our model-adaptive data augmentations can bring more improvement to their representation invariance and loss smoothness. In Fig. 5, we conduct a thorough sensitivity analysis on $\tau\%$ over three tasks and find that sample selection with TCS can effectively improve the performance. Moreover, in this way, we do not need to apply our augmentation to every training samples and thus save the training cost.

### A.2  Fast Lagaragian Attack Method

We use the fast lagaragian perceptual attack method (Algorithm 3 in [24]) to solve the Lagragian multiplier function in Equation.(5), which finds the optimal $x'$ through gradient descent over $x'$, starting at $x$ with a small amount of noise added. During the $T$ gradient descent steps, $\lambda$ is increased exponentially form 1 to 10 and the step size is decreased. $T$ is set to be 5 for all the experiments.

---

**Algorithm 2** Fast Lagarangian Attack Method

---

**Input:**
    Training data $(x,y)$; The class preserving margin $\sigma$; Neural Network $F(\cdot)$
**Output:**
1:  $x' = x + 0.01 * \mathcal{N}(0, 1)$
2: **for** $t = 1, ..., T$ **do**
3:    $\lambda \leftarrow 10^{t/T}$
4:    $\triangle = -\nabla_{x'}[\|\phi(x) - \phi(x')\|_2 - \lambda\max(0, \log F(x;\theta)[y] - \log F(x';\theta)[y] - \sigma)]$
5:    $\hat{\triangle} = \triangle/\|\triangle\|_2$
6:    $\gamma = \epsilon * (0.1)^t/T$
7:    $m \leftarrow (F(x;\theta)[y] - F(x' + h\hat{\triangle};\theta)[y])/h$
8:    $x' \leftarrow x + (\gamma/m)\hat{\triangle}$
9: **end for**

---

## B  Additional Theoretical Results and Proofs

### B.1  Proof of Theorem 4.2

***Proof of Theorem 4.2.*** Problem (1) contains two versions of objectives for $\mathbf{Z}'$:

$$\text{argmax}_{\mathbf{Z}'} I(\mathbf{Z}' \wedge \mathbf{X}') \text{ subject to } I(\mathbf{Z}' \wedge \mathbf{A}) = 0, \tag{8}$$

or

$$\operatorname{argmax}_{\mathbf{Z}'} I(\mathbf{Z}' \wedge \mathbf{Y}) \text{ subject to } I(\mathbf{Z}' \wedge \mathbf{A}) = 0. \tag{9}$$

Both Problem (8) and Problem (9) lead to the $\epsilon$-minimal sufficient representation $\mathbf{Z}^*$. We first prove the more challenging Problem (8) objective.

**I. For Problem** (8)**:** $\operatorname{argmax}_{\mathbf{Z}'} I(\mathbf{Z}' \wedge \mathbf{X}')$ subject to $I(\mathbf{Z}' \wedge \mathbf{A}) = 0$.

We first prove the sufficiency of $\mathbf{Z}^*$, then prove the $\epsilon$-minimality of $\mathbf{Z}^*$.

### 1) Proof of sufficiency

Since $I(\mathbf{Z}' \wedge \mathbf{X}') = H(\mathbf{X}') - H(\mathbf{X}'|\mathbf{Z}')$, and $H(\mathbf{X}')$ does not depend on $\mathbf{Z}'$, we have that the solution to Problem (8), $\mathbf{Z}^*$, minimizes $H(\mathbf{X}'|\mathbf{Z}')$ under constraint $I(\mathbf{Z}' \wedge \mathbf{A}) = 0$.

Then, we show that $\mathbf{Z}^*$ also minimizes $H(\mathbf{X}|\mathbf{Z}')$.

We know that $I(\mathbf{X}, \mathbf{A} \wedge \mathbf{Z}'|\mathbf{X}') = 0$ because of the Markovian property. Since $I(\mathbf{X}, \mathbf{A} \wedge \mathbf{Z}'|\mathbf{X}') = H(\mathbf{X}, \mathbf{A}|\mathbf{X}') - H(\mathbf{X}, \mathbf{A}|\mathbf{X}', \mathbf{Z}')$, we have

$$H(\mathbf{X}, \mathbf{A}|\mathbf{X}') = H(\mathbf{X}, \mathbf{A}|\mathbf{X}', \mathbf{Z}'). \tag{10}$$

Then we can derive

$$H(\mathbf{X}, \mathbf{A}|\mathbf{Z}') - H(\mathbf{X}, \mathbf{A}|\mathbf{X}') = H(\mathbf{X}, \mathbf{A}|\mathbf{Z}') - H(\mathbf{X}, \mathbf{A}|\mathbf{X}', \mathbf{Z}') \tag{11}$$

$$= I(\mathbf{X}, \mathbf{A} \wedge \mathbf{X}'|\mathbf{Z}') \tag{12}$$

$$= H(\mathbf{X}'|\mathbf{Z}') - H(\mathbf{X}'|\mathbf{Z}', \mathbf{X}, \mathbf{A}) \tag{13}$$

$$= H(\mathbf{X}'|\mathbf{Z}') \tag{14}$$

Equality (13) holds because $\mathbf{X}'$ comes from a deterministic function of $\mathbf{X}$ and $\mathbf{A}$. Since $H(\mathbf{X}, \mathbf{A}|\mathbf{X}')$ does not depend on $\mathbf{Z}'$, $\mathbf{Z}^*$ minimizes $H(\mathbf{X}, \mathbf{A}|\mathbf{Z}')$ as it minimizes $H(\mathbf{X}'|\mathbf{Z}')$.

Also, we known that $I(\mathbf{Z}^* \wedge \mathbf{A}) = 0$, so we can further obtain

$$H(\mathbf{X}'|\mathbf{Z}') = H(\mathbf{X}, \mathbf{A}|\mathbf{Z}') \tag{15}$$

$$= H(\mathbf{X}|\mathbf{Z}') + H(\mathbf{A}|\mathbf{Z}') - I(\mathbf{X} \wedge \mathbf{A}|\mathbf{Z}') \tag{16}$$

$$= H(\mathbf{X}|\mathbf{Z}') + H(\mathbf{A}|\mathbf{Z}') - H(\mathbf{A}|\mathbf{Z}') + H(\mathbf{A}|\mathbf{X}, \mathbf{Z}') \tag{17}$$

$$= H(\mathbf{X}|\mathbf{Z}') + H(\mathbf{A}|\mathbf{X}, \mathbf{Z}') \tag{18}$$

$$= H(\mathbf{X}|\mathbf{Z}'), \tag{19}$$

where Equation (19) holds because $H(\mathbf{A}|\mathbf{X}, \mathbf{Z}') \leq H(\mathbf{A}|\mathbf{Z}') = 0$.

Therefore, $\mathbf{Z}^*$ minimizes $H(\mathbf{X}|\mathbf{Z}')$.

Following the similar procedure as above (Equation (10) to Equation (19)), we are able to show that

$$H(\mathbf{X}|\mathbf{Z}') = H(\mathbf{Y}, \mathbf{N}|\mathbf{Z}') - H(\mathbf{Y}, \mathbf{N}|\mathbf{X}) \tag{20}$$

So $\mathbf{Z}^*$ also minimizes $H(\mathbf{Y}, \mathbf{N}|\mathbf{Z}')$, which can be further decomposed into $H(\mathbf{Y}|\mathbf{Z}') + H(\mathbf{N}|\mathbf{Z}', \mathbf{Y})$. Next we show by contradiction that $H(\mathbf{Y}|\mathbf{Z}^*)$ equals to $H(\mathbf{Y}|\mathbf{X})$ and thus $I(\mathbf{Y} \wedge \mathbf{Z}^*) = I(\mathbf{Y} \wedge \mathbf{X})$.

Define $L(\mathbf{Z}') := H(\mathbf{Y}|\mathbf{Z}') + H(\mathbf{N}|\mathbf{Z}', \mathbf{Y})$ Assume that the optimizer $\mathbf{Z}^*$ minimizes $L$, but does not satisfy sufficiency, i.e., $H(\mathbf{Y}|\mathbf{Z}^*) > H(\mathbf{Y}|\mathbf{X}) = 0$. We will then show that one can construct another representation $\hat{\mathbf{Z}}$ such that $L(\hat{\mathbf{Z}}) < L(\mathbf{Z}^*)$, conflicting with the assumption that $\mathbf{Z}^*$ minimizes $L$. The construction of $\hat{\mathbf{Z}}$ works as follows. Since the augmented data $\mathbf{X}'$ satisfies $I(\mathbf{X}' \wedge \mathbf{Y}) = I(\mathbf{X} \wedge \mathbf{Y})$ (Condition (a) of Theorem 4.2), we have $H(\mathbf{Y}|\mathbf{X}') = H(\mathbf{Y}|\mathbf{X}) = 0$. Hence, there exists a function $\pi'$ such that $\pi'(\mathbf{X}') = \mathbf{Y}$. Define $\hat{\mathbf{Z}} := (\mathbf{Z}^*, \pi(\mathbf{X}'))$, then we have

$$L(\hat{\mathbf{Z}}) = H(\mathbf{Y}|\hat{\mathbf{Z}}) + H(\mathbf{N}|\hat{\mathbf{Z}}, \mathbf{Y}) \tag{21}$$

$$= H(\mathbf{Y}|\mathbf{Z}^*, \pi(\mathbf{X}')) + H(\mathbf{N}|\mathbf{Z}^*, \pi(\mathbf{X}'), \mathbf{Y}) \tag{22}$$

$$= H(\mathbf{Y}|\mathbf{Z}^*, \mathbf{Y}) + H(\mathbf{N}|\mathbf{Z}^*, \mathbf{Y}, \mathbf{Y}) \tag{23}$$

$$= 0 + H(\mathbf{N}|\mathbf{Z}^*, \mathbf{Y}) \tag{24}$$

$$< H(\mathbf{Y}|\mathbf{Z}^*) + H(\mathbf{N}|\mathbf{Z}^*, \mathbf{Y}) \tag{25}$$

$$= L(\mathbf{Z}^*) \tag{26}$$

Therefore, the constructed $\hat{\mathbf{Z}}$ conflicts with the assumption. We can conclude that any optimizer $\mathbf{Z}^* \in \operatorname{argmin}_{\mathbf{Z}'} H(\mathbf{Y}|\mathbf{Z}') + H(\mathbf{N}|\mathbf{Z}', \mathbf{Y})$ has to satisfy $H(\mathbf{Y}|\mathbf{Z}^*) = H(\mathbf{Y}|\mathbf{X})$, which is equivalent to $I(\mathbf{Y} \wedge \mathbf{Z}^*) = I(\mathbf{Y} \wedge \mathbf{X})$. The sufficiency of $\mathbf{Z}^*$ is thus proven.

As a result, the maximizer to Problem (8), $\mathbf{Z}^*$, satisfies $I(\mathbf{Z}^* \wedge \mathbf{Y}) = I(\mathbf{X}' \wedge \mathbf{Y}) = I(\mathbf{X} \wedge \mathbf{Y})$.

### 2) Proof of $\epsilon$-Minimality

Since $\mathbf{X}$ is a deterministic function of $\mathbf{Y}$ and $\mathbf{N}$, we have

$$I(\mathbf{X}' \wedge \mathbf{X}) = I(\mathbf{X}' \wedge \mathbf{Y}, \mathbf{N}) \tag{27}$$

$$= I(\mathbf{X}' \wedge \mathbf{N}) + I(\mathbf{X}' \wedge \mathbf{Y}|\mathbf{N}) \tag{28}$$

$$\leq I(\mathbf{X}' \wedge \mathbf{Y}|\mathbf{N}) + \epsilon \tag{29}$$

where the equality in Equation (27) holds because $I(\mathbf{X}' \wedge \mathbf{X}) \geq I(\mathbf{X}' \wedge \mathbf{Y}, \mathbf{N})$ and $I(\mathbf{X}' \wedge \mathbf{X}) \leq I(\mathbf{X}' \wedge \mathbf{Y}, \mathbf{N})$ both hold.

And we can derive

$$\begin{aligned}
I(\mathbf{X}' \wedge \mathbf{Y}|\mathbf{N}) - I(\mathbf{X} \wedge \mathbf{Y}) &= H(\mathbf{Y}|\mathbf{N}) - H(\mathbf{Y}|\mathbf{X}', \mathbf{N}) - H(\mathbf{Y}) + H(\mathbf{Y}|\mathbf{X}) \\
&= H(\mathbf{Y}|\mathbf{X}) - H(\mathbf{Y}|\mathbf{X}', \mathbf{N}) \\
&\leq H(\mathbf{Y}|\mathbf{X}) \\
&= 0
\end{aligned} \tag{30}$$

Moreover, we know that

$$I(\mathbf{Y} \wedge \mathbf{N}) + I(\mathbf{X}' \wedge \mathbf{Y}|\mathbf{N}) = I(\mathbf{X}' \wedge \mathbf{Y}) + I(\mathbf{Y} \wedge \mathbf{N}|\mathbf{X}') \tag{31}$$

And we have $I(\mathbf{Y} \wedge \mathbf{N}) = 0$, so

$$\begin{aligned}
I(\mathbf{X}' \wedge \mathbf{Y}|\mathbf{N}) - I(\mathbf{X} \wedge \mathbf{Y}) &= I(\mathbf{X}' \wedge \mathbf{Y}|\mathbf{N}) - I(\mathbf{X}' \wedge \mathbf{Y}) \\
&= I(\mathbf{Y} \wedge \mathbf{N}|\mathbf{X}') \geq 0
\end{aligned} \tag{32}$$

Combining (30) and (32), we have

$$I(\mathbf{X}' \wedge \mathbf{Y}|\mathbf{N}) = I(\mathbf{X} \wedge \mathbf{Y}) \tag{33}$$

Note that (33) holds for all sufficient statistics of $\mathbf{X}$ w.r.t. $\mathbf{Y}$.

Then we first show that $\mathbf{X}'$ is $\epsilon$-minimal of $\mathbf{X}$ w.r.t. $\mathbf{Y}$ by contradiction.

Assume there exists a random variable $\tilde{\mathbf{X}}$ satisfying $I(\tilde{\mathbf{X}} \wedge \mathbf{Y}) = I(\mathbf{X} \wedge \mathbf{Y})$, such that $I(\tilde{\mathbf{X}} \wedge \mathbf{X}) < I(\mathbf{X}' \wedge \mathbf{X}) - \epsilon$.

Then we have

$$I(\tilde{\mathbf{X}} \wedge \mathbf{Y}|\mathbf{N}) = I(\tilde{\mathbf{X}} \wedge \mathbf{X}) - I(\tilde{\mathbf{X}} \wedge \mathbf{N}) \tag{34}$$

$$\leq I(\tilde{\mathbf{X}} \wedge \mathbf{X}) \tag{35}$$

$$< I(\mathbf{X}' \wedge \mathbf{X}) - \epsilon \tag{36}$$

$$= I(\mathbf{X}' \wedge \mathbf{Y}|\mathbf{N}) + \epsilon - \epsilon \tag{37}$$

$$= I(\mathbf{X} \wedge \mathbf{Y}) \tag{38}$$

$$= I(\tilde{\mathbf{X}} \wedge \mathbf{Y}|\mathbf{N}) \tag{39}$$

where (34) holds by replacing $\mathbf{X}'$ with $\tilde{\mathbf{X}}$ in (28).

Hence, we get $I(\tilde{\mathbf{X}} \wedge \mathbf{Y}|\mathbf{N}) < I(\tilde{\mathbf{X}} \wedge \mathbf{Y}|\mathbf{N})$ which is impossible. So we have that there does not exist such an $\tilde{\mathbf{X}}$, and $\mathbf{X}'$ is $\epsilon$-minimal representation of $\mathbf{X}$ w.r.t. $\mathbf{Y}$.

Then, since we have $I(\mathbf{Z}' \wedge \mathbf{N}) \leq I(\mathbf{X}' \wedge \mathbf{N}) \leq \epsilon$ (thanks to Data Processing Inequality), $\mathbf{Z}'$ is also a $\epsilon$-minimal sufficient statistic of $\mathbf{X}$ w.r.t. $\mathbf{Y}$.

**II. For Problem** (9): $\mathrm{argmax}_{\mathbf{Z}'} I(\mathbf{Z}' \wedge \mathbf{Y})$ subject to $I(\mathbf{Z}' \wedge \mathbf{A}) = 0$.

Since the objective is to maximize $I(\mathbf{Z}' \wedge \mathbf{Y})$, we only need to show that $\mathbf{Z}^*$ achieves the maximum mutual information with $\mathbf{Y}$. According to the above proof for Problem (8), we know that there exist $\mathbf{Z}'$ such that $I(\mathbf{Z}' \wedge \mathbf{A}) = 0$ and $I(\mathbf{Z}' \wedge \mathbf{Y}) = I(\mathbf{X} \wedge \mathbf{Y})$. Hence, the optimizer to Problem (9) must satisfy sufficiency.

The proof of $\epsilon$-minimality is identical to the one under Problem (8).

$\square$

## B.2  Additional Theoretical Results on Augmentation Properties

The two conditions in Theorem 4.2, Condition (a) or Condition (b), requires that the augmentation $\mathbf{X}'$ is (a) sufficient and (b) ($\epsilon$)-minimal. These two conditions are closely related to some augmentation rationales in prior papers. For example, Wang et al. [45] propose a **symmetric augmentation**, which can result in Condition (a), as formalized in Lemma B.1 below. Furthermore, Tian et al. [40] propose an "InfoMin" principle of data augmentation, that minimizes the mutual information between different views (equivalent to $\min I(\mathbf{X}, \mathbf{X}')$). We show by Lemma B.2 that this InfoMin principle leads to the above Condition (b). In contrast, our Theorem 4.2 characterizes two key conditions of augmentation and directly relate them to the optimality of the learned representation.

**Lemma B.1** (Sufficiency of Augmentation). *Suppose the original and augmented observations* $\mathbf{X}$ *and* $\mathbf{X}'$ *satisfy the following properties:*

$$P(\mathbf{X} = u, \mathbf{X}' = v|\mathbf{Y} = y) = P(\mathbf{X} = v, \mathbf{X}' = u|\mathbf{Y} = y), \ \forall u, v, y \tag{40a}$$

$$P(\mathbf{X} = u|\mathbf{Y} = y) = P(\mathbf{X}' = u|\mathbf{Y} = y) \tag{40b}$$

*Then the augmented observation* $\mathbf{X}'$ *is sufficient for the label* $\mathbf{Y}$, *i.e.,* $I(\mathbf{X}' \wedge \mathbf{Y}) = I(\mathbf{X} \wedge \mathbf{Y})$.

*Proof of Lemma B.1.*

$$I(\mathbf{X}' \wedge \mathbf{Y}) = \sum_{x,y} P(\mathbf{X}' = x, \mathbf{Y} = y) \log \frac{P(\mathbf{X}') = x, \mathbf{Y} = y)}{P(\mathbf{X}' = x)P(\mathbf{Y} = y)} \tag{41}$$

$$= \sum_{x,y} P(\mathbf{X}' = x|\mathbf{Y} = y)P(\mathbf{Y} = y) \log \frac{P(\mathbf{X}' = x|\mathbf{Y} = y)}{\sum_{\bar{y}} P(\mathbf{X}' = x|\mathbf{Y} = \bar{y})P(\mathbf{Y} = \bar{y})} \tag{42}$$

$$= \sum_{x,y} P(\mathbf{X} = x|\mathbf{Y} = y)P(\mathbf{Y} = y) \log \frac{P(\mathbf{X} = x|\mathbf{Y} = y)}{\sum_{\bar{y}} P(\mathbf{X} = x|\mathbf{Y} = \bar{y})P(\mathbf{Y} = \bar{y})} \tag{43}$$

$$= \sum_{x,y} P(\mathbf{X} = x, \mathbf{Y} = y) \log \frac{P(\mathbf{X}) = x, \mathbf{Y} = y)}{P(\mathbf{X} = x)P(\mathbf{Y} = y)} \tag{44}$$

$$= I(\mathbf{X} \wedge \mathbf{Y}) \tag{45}$$

where the third equation utilizes the property of symmetric augmentation. $\square$

**Lemma B.2** (Maximal Insensitivity to Nuisance). *If Assumption 4.1 holds, i.e.,* $H(\mathbf{Y}|\mathbf{X}) = 0$, *the mutual information* $I(\mathbf{X}' \wedge \mathbf{X})$ *can be decomposed as*

$$I(\mathbf{X}' \wedge \mathbf{X}) = I(\mathbf{X}' \wedge \mathbf{N}) + I(\mathbf{X}' \wedge \mathbf{Y}) \tag{46}$$

*Since* $\mathbf{X}'$ *is sufficient, i.e.,* $I(\mathbf{X}' \wedge \mathbf{X}) = I(\mathbf{X} \wedge \mathbf{Y})$ *is a constant, minimizing* $I(\mathbf{X}' \wedge \mathbf{X})$ *is equivalent to minimizing* $I(\mathbf{X}' \wedge \mathbf{N})$.

Lemma B.2 can be obtained by a simple adaptation from Proposition 3.1 by Achille and Soatto [1].

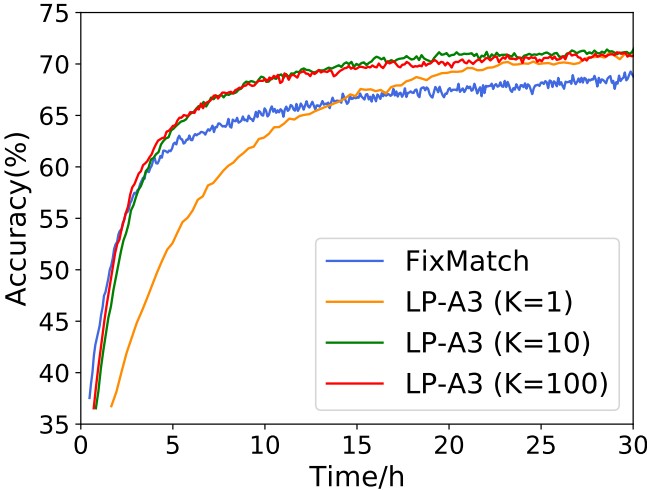

Figure 6: **Walk-clock time comparison** on CIFAR100 with 2500 labeled data.

## C  Experiments

### C.1  Implementation Details

All codes are implemented with Pytorch[1]. To train the neural net with LP-A3 augmentation, we apply seperate batch norm layer (BN), i.e., agumented data and normal data use different BN, which is a common strategy used by previous adversarial augmentations [19, 48, 51]. The only hyperparameters for LP-A3 are label preserving margin $\sigma$ and data selection ratio $\tau$, which are tuned for each task according to the results in Sec.6.3.

**Semi-supervised learning**  We reproduce Fixmatch [36] based on public code[2] and apply LP-A3 to it. Following [36], we used a Wide-ResNet-28-2 with 1.5M parameters for CIFAR10, WRN-28-8 for CIFAR100, and WRN-37-2 for STL-10. All the models are trained for $2^{18}$ iterations. $\sigma$ is set to 0.002 for CIFAR10 and STL-10, and 0.02 for CIFAR100 and $\tau$ is set to be 90. Since FixMatch only apply data augmentation to those unlabeled data, here LP-A3 is also applied to those unlabeled data as data augmentation. For unlabeled data, label $\mathbf{Y}$ used in LP-A3 is the pesudo label generated by FixMatch algorithm.

**Noisy-label learning**  We reproduce DivideMix [26] and PES [5] based on their official code[3] and apply LP-A3 to them as data augmentation. Following [26, 5], we used a ResNet-18 for CIFAR10 and CIFAR100. All the models are trained for 300 epochs. $\sigma$ is set to 0.002 for CIFAR10 and 0.02 for CIFAR100, and $\tau$ is set to be 90. All the noise are symmetric noise. For noisy labeled data, label $\mathbf{Y}$ used in LP-A3 is the pesudo label generated by DivideMix or PES algorithm respectively.

**Medical Image Classification**  Here we follow the original training and evaluation protocol of MedMNIST [4] and apply LP-A3 to the training procedure as data augmentation. ResNet-18 and ResNet-50 are trained for 100 epochs with cross-entropy loss on all the multi-class classfication subset of MedMNIST. $\sigma$ is set to 0.02 and $\tau$ is tuned from $\{20, 50, 90\}$ for each dataset. The hyperparameters of RandAugment [13] is set to $N = 3, M = 5$ by following their original paper.

**Sensitivity Analysis of Hyperparameters**  In Figure. 5, the experiments for semi-supervised learning are conducted on CIFAR100 with 2500 labeled data, the experiments for noisy-label learning are conducted on CIFAR100 with 80% noisy label, and the experiments for medical image classification are conducted on DermaMNIST with ResNet50.

---

[1]https://pytorch.org/

[2]https://github.com/kekmodel/FixMatch-pytorch

[3]https://github.com/LiJunnan1992/DivideMix, https://github.com/tmllab/PES

[4]https://github.com/MedMNIST/experiments

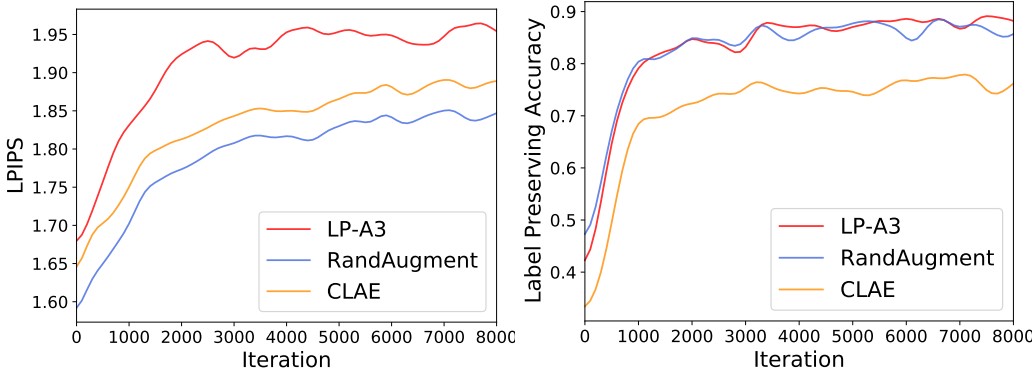

(a) $-I(\mathbf{X}' \wedge \mathbf{X})$ meadured by LPIPS distance     (b) $I(\mathbf{X}' \wedge \mathbf{Y})$ measured by classification accuracy on $x'$

Figure 7: **Mutual information terms** of LP-A3, RandAugment and CLAE during training.

Table 4: Semi-supervised Learning performance on ImageNet. We eualate the performance of FixMatch and "FixMatch+LP-A3" at different training stages. Iter denotes training iterations.

|  | 100,000 Iter | 250,000 Iter | 400,000 Iter |
| --- | --- | --- | --- |
| FixMatch | 44.45 | 58.79 | 62.87 |
| FixMatch+LP-A3 | 51.22 | 60.15 | 63.67 |

## C.2    Computational Cost

Although Algorithm 2 is a pretty fast algorithm to solve Lagragian multiplier function, it still requires several gradient descent steps which is computationally expensive. One way to reduce computational cost is to generate LP-A3 for every few epochs. To be specific, once LP-A3 is generated, it will be saved and used to train the network for the next $K$ epochs. When $K = 1$, it degenerates to the original Algorithm 1. The walk-clock time comparison on CIFAR100 with 2500 labeled data is give in Fig. 6, where we can see that LP-A3 ($K = 10$) and LP-A3 ($K = 100$) achieves much better accuracy than baseline within the same training time. Moreover, LP-A3 ($K = 10$) and LP-A3 ($K = 100$) achieves comparable accuracy as the original LP-A3 ($K = 1$) after convergence, indicating that LP-A3 is quite informative and it takes several epochs for the neural net to learn from it.

## C.3    Mutual Information Terms of Different Data Augmentation

In order to further analyze the properties of different data augmentations, here we report the value of mutual information terms $I(\mathbf{X}' \wedge \mathbf{X})$ and $I(\mathbf{X}' \wedge \mathbf{Y})$ of training data generated by LP-A3, RandAugment and CLAE [19] (an adversarial augmentation method) during the training procedure. The results on CIFAR10 are given in Table. 7, where $-I(\mathbf{X}' \wedge \mathbf{X})$ is measured by the LPIPS distance between $x$ and $x'$ and $I(\mathbf{X}' \wedge \mathbf{Y})$ is measured by the label preserving accuracy, i.e., the classification accuracy of the current model on $x'$. We can clearly see that LP-A3 is the most different from the original data (largest LPIPS distance) and at the same time preserves the label well. Although RandAugment can also preserve label, as a pre-defined augmentation, it is the closest to the original data. Another adaptive augmentation CLAE has larger LPIPS ditance than RandAugment but cannot preserve the label well, achieving the lowest label preserving accuracy.

## C.4    ImageNet Experiments

In order to validate the performance of LP-A3 on large-scale datasets, we compare FixMatch and "FixMatch+LP-A3" on ImageNet the semi-supervised learning. ResNet-50 is used as the backbone network. We randomly choose 10% data and set them as labeled data, while the remaining 90% are unlabeled data. The batch size for labeled (unlabeled) data is 64 (320). The results are reported in the Table 4, in which LP-A3 can improve FixMatch by a large margin, especially during the early stage (>6% at 100,000 iterations).

Table 5: Comparison with Mixup and Adversarial AutoAugment on the full dataset of CIFAR10.

| CIFAR10 | 50 epochs | 100 epochs | 150 epochs |
|---|---|---|---|
| Mixup | 87.39 | 93.74 | 96.05 |
| Mixup+LPA3 | 86.82 | 94.73 | 96.12 |
| Adversarial autoaugment | 89.70 | 95.05 | 97.09 |
| Adversarial autoaugment+LPA3 | 90.45 | 95.33 | 97.20 |

## C.5 Comparison with Mixup and Adversarial AutoAugment

In this section, we compare LP-A3 with state-of-the-art data augmentation on the full dataset in a supervised manner. We select Mixup [56] and Adversarial AutoAugment [60] as the representative of sample-mixing based augmentaiton and automated data augmentation methods respectively, and we empirically compare LP-A3 with them by applying LP-A3 on top of them. The backbone model is WideResNet-28-10 and all the model are trained for 150 epochs. The results are given in Table 5. It shows that Mixup and Adversarial Autoaugment achieves very high accuracy (>96%) on CIFAR10 as they are designed for this task, but LP-A3 can further improve them, which indicates that LP-A3 is complementaray to these previous data augmentation using domain knowledge. Note in early stage the advantage of LP-A3 over Adversarial Autoaugment is especially significant ($> 0.7\%$).