# OpenReview forum: "Adversarial Auto-Augment with Label Preservation: A Representation Learning Principle Guided Approach"
_NeurIPS.cc/2022/Conference — NeurIPS 2022 Accept_

### Official Review · Reviewer_VALt · 2022-07-04

**Rating:** 5
**Confidence:** 3
**Soundness:** 3 good
**Presentation:** 3 good
**Contribution:** 2 fair

**Summary:**

This paper proposes a new data augmentation method that does not require training an extra generative model but instead leverages the intermediate layer representations of the end-task model for generating data augmentations. The augmentation is built based on a representation learning principle that aims to preserve the minimum sufficient information of the labels. Experiments across different datasets are conducted to demonstrate the effectiveness of the proposed method.

After rebuttal:
Thanks for the authors' responses. The authors address some of my concerns so I raise my rate.

**Questions:**

Please refer to the weaknesses.

**Limitations:**

-

**Strengths And Weaknesses:**

Strengths

+ A novel, simple, and intuitive idea for data augmentation
+ The simple idea is very effective when implemented correctly
+ The paper is easy to follow, well-written, and inspiring. Moreover, the theoretical interpretation of the method is sound.
+ Results with various different tasks, including semi-supervised learning, noisy-label learning, and medical image classification, and datasets are provided in this manuscript to show the effectiveness of the proposed LP-A3.

Weakness
- How does the TCS select the hard positive data in Alg.1? I think it is necessary to provide the selection detail to make the manuscript self-contained.
- The experimental results in the manuscript are insufficient to show the effectiveness of LP-A3. In semi-supervised learning tasks, only one augmentation method, InfoMin, is tested on STL-10 dataset as a comparison. What are the backbone networks in the experiments in Table 2? Moreover, How’s the performance of LP-A3 on CIFRA10/100 in a supervised classification manner? Evaluation with other backbone networks rather than ResNet-18 and ResNet-50 is also important.
- How’s the performance of LP-A3 on the large-scale dataset, such as ImageNet
- I’m curious about the results in Fig.5. As the proposed LP-A3 is designed to generate the augmented X’ to preserve the minimum sufficient information of the label, why the performance of LP-A3 can drop when all data is selected for augmentation? The author stated that some data augmentations are useless to apply data augmentation, which did not convince me.

---

> ### Author Response · Authors · 2022-08-02
> **To Reviewer VALt: Part I**
>
> We appreciate your time and suggestions! We have added new experiments on ImageNet and comparisons to other data augmentations in the new version of our paper. Here are our detailed replies to your questions.
>
> **(1) How does the TCS select the hard positive data in Alg.1? I think it is necessary to provide the selection detail to make the manuscript self-contained.**
>
> TCS[1] measures the consistency of the model output (i.e., the predicted class distribution) for a sample over historical training steps. We select the top $\tau$\% sample with the lowest TCS to apply our data augmentation because samples with small TCS tend to have sharp loss landscapes. These samples provide more informative gradients than others and applying our model-adaptive data augmentations can bring more improvement to their representation invariance and loss smoothness. In Figure (5), we conduct a thorough sensitivity analysis on $\tau\%$ over three tasks and find that sample selection with TCS can effectively improve the performance. Moreover, in this way, we do not need to apply our augmentation to every training sample and thus save the training cost. We have added details about TCS in Sec.B of the Appendix.
>
> **(2) In semi-supervised learning tasks, only one augmentation method, InfoMin, is tested on STL-10 dataset as a comparison.**
>
> - As the baseline of learnable data augmentation for semi-supervised learning, InforMin is the only published work we could find.
> - For hand-crafted data augmentation, FixMatch uses RandAugment and CTAuemgnt which are specifically selected or designed for semi-supervised learning. As a general auto-augmentation technique not designed for a specific application only, LP-A3 does not rely on any human expert knowledge but can achieve further improvement on top of existing augmentations. This demonstrates its superiority and advantages.
>
> **(3) Moreover, How’s the performance of LP-A3 on CIFRA10/100 in a supervised classification manner?**
>
> - A great number of methods are mainly designed and tuned for the fully supervised learning on CIFAR10 and CIFAR100, leading to overfitting and poor true generalization performance as discussed in [2], especially for the state-of-the-art methods on these datasets. In this paper, we instead focus on those tasks deficient in or heavily relying on high-quality data augmentations, e.g., semi-supervised/noisy-label learning which lacks labeled or correctly labeled data, medical images in MedMNIST that lack sufficient domain knowledge.
> - Our method can be easily implemented for these three diverse and important categories of learning tasks and consistently improve all their performance --- This is a non-trivial contribution because in previous work every task needs to tune its own augmentation operations to reach the best performance, e.g., SimCLR and FixMatch. There are so many possible tasks in the world and we cannot expect that one single augmentation technique can achieve the best performance on all of them.
> - LPA3 does not aim at replacing existing hand-crafted data augmentation but are complementary to them.
> - We select Mixup and Adversarial AutoAugment as the representative of sample-mixing based augmentation and automated data augmentation respectively, and we empirically compare LP-A3 with them by applying LP-A3 on top of them. The results are reported in the table below, which shows that Mixup and Adversarial Autoaugment achieves high accuracy (>96\%) on CIFAR10 as they are designed for this task, but LP-A3 can further improve them, which indicates that LP-A3 are complementary to these previous data augmentations designed using domain knowledge. Note in the early stage the advantage of LP-A3 over Adversarial Autoaugment is especially significant (>0.7\%). These additional results have been added to Section D of the Appendix in the lately revised version.
>
> CIFAR10|50 epochs|100 epochs|150 epochs|
> ---|:--:|---:|---:
> Mixup|87.39|93.74|96.05
> Mixup+LPA3|86.82|94.73|96.12
> Adversarial Autoaugment|89.70|95.05|97.09
> Adversarial Autoaugment+LPA3|90.45|95.33|97.20
>
> **(4) How’s the performance of LP-A3 on the large-scale dataset, such as ImageNet.**
>
> In order to validate the performance of LP-A3 on large-scale datasets, we compare FixMatch and "FixMatch+LP-A3" on ImageNet the semi-supervised learning. ResNet-50 is used as the backbone network. We randomly choose 10\% data and set them as labeled data, while the remaining 90% are unlabeled data. The results are reported in the table below, in which LP-A3 can improve FixMatch by a large margin, especially during the early stage (>6$\%$ at 100,000 iterations).  These additional results have been added to Section D of the Appendix in the revised version.
>
> ImageNet|100k Iter|250k Iter|400k Iter
> ---|:--:|---:|---:
> FixMatch|44.45|58.79|62.87
> FixMatch+LP-A3|51.22|60.15|63.67

---

> > ### Author Response · Authors · 2022-08-02
> > **Response to Reviewer VALt: Part II**
> >
> > **(5) What are the backbone networks in the experiments in Table 2? Evaluation with other backbone networks rather than ResNet-18 and ResNet-50 is also important.**
> >
> > We follow the previous work and use the standard backbone networks for each setting. Details can be found in Section D of the Appendix.
> > - In Table 1,  following FixMatch, we use WideResNet28-2, WideResNet28-8 and  WideResNet37-2 for CIFAR10, CIFAR100 and STL10 respectively.
> > - In Table 2, following DivideMix, we use ResNet18 for CIFAR10 and CAFAR100.
> > - In Table 3, following MedMNIST, we use ResNet18 and ResNet50.
> >
> > As a result, the evaluation in this paper has covered various kinds of models.
> >
> > **(6) Why the performance of LP-A3 can drop when all data is selected for augmentation?**
> >
> > Those data with high time consistency have smooth and flat loss landscapes and are far away from the label-preserving boundary. As a result, LP-A3 allows a large perturbation to those data under the label-preserving constraint, making the resulted augmentations drift away from the original image and its local manifold so the augmentations can be out-of-distribution data. Training with those out-of-distribution data may degenerate the model. Hence, we avoid applying LP-A3 to those data with high time consistency.
> >
> > ---
> > References:
> >
> > [1] Tianyi Zhou, Shengjie Wang, and Jeff A. Bilmes. Time-Consistent Self-Supervision for Semi-Supervised Learning. International Conference on Machine Learning (ICML), 2020.
> >
> > [2] Benjamin Recht, Rebecca Roelofs, Ludwig Schmidt and Vaishaal Shankar. Do CIFAR-10 Classifiers Generalize to CIFAR-10?

---

### Official Review · Reviewer_JwsS · 2022-07-06

**Rating:** 5
**Confidence:** 3
**Soundness:** 2 fair
**Presentation:** 3 good
**Contribution:** 2 fair

**Summary:**

The paper proposes a data augmentation objective, LP-A3,  to create label-preserving adversarial examples that improve the model generalization of deep learning models. More precisely, the method perturbs an input image to maximize its perceptual difference while constraining its predicted class probability to be similar to the original image. Experimental results show the effectiveness of the proposed objective in supervised, semi-supervised and noisy-label learning tasks.

**Questions:**

__O1.__ It seems that LP-A3 relies on the representation learning network $F$ to compute the LPIPS distance $\phi(x)$ and softmax class probability $F(x; \theta)[y]$ in Eq. (4). During the early stages of training, when $F$ is not well trained, is it possible that it may generate incorrect training signals for computing the LPIPS distance and softmax class probability for the label-preserving constrain? How does the method deal with this problem especially for small and more complicated datasets?

**Limitations:**

Please see the above concerns.

**Strengths And Weaknesses:**

__Strengths:__

__S1. [reasonable and well-motivated]__ Data augmentation is crucial for modern deep learning methods and becomes a bottleneck when deploying them to domains with less prior knowledge. From this perspective, the paper is tackling an important problem.

__S2. [practical solution]__ The method is easy to be integrated into existing learning frameworks.

__S3. [clear writing]__ Overall, the paper is easy to follow.

__Weakness:__

__W1. [Missing related work]__ There are some related works this paper omitted. For example, Adversarial AutoAugment [1] also uses an adversarial objective to learn an augmentation strategy formed by a set of label-preserving transformations for the target dataset. The paper can compare LP-A3 with other adversarial data augmentation strategies in terms of their idea and performance, so we will know which approach is better in which scenarios.

__W2. [Insufficient experiment]__

1. The experiments are not extensive enough. Only the results on several tiny datasets are provided. The results on the large-scale ImageNet dataset will make this paper stronger. In supervised image classification tasks, the paper compares LP-A3 with RandAugment on MedMNIST. It would be better to compare LP-A3 with other Automated Data Augmentation methods, like AutoAugment, Adversarial AutoAugment, and sample-mixing augmentations, like Mixup, Cutmix, on the full dataset of popular classification benchmarks, e.g. CIFAR10 and CIFAR100. This helps to verify the performance gains compared with the state-of-the-art data augmentation schemes.

2. Prior works such as RandAugment run multiple times for the model and present the mean and standard deviation. Did the authors run multiple times for each model and what about presenting the mean and standard deviation?

__W3. [Adversarial training]__ According to lines 252-255 and Appendix B, it seems that LP-A3 uses an adversarial-attack-like procedure to perturb the input images.  Adversarial attack perturbs an image to alter the class label; oppositely, LP-A3 aims to preserve the class label. With the opposing objectives, can LP-A3 be integrated with adversarial training? Does LP-A3 affect the use of perturbation-based adversarial attacks to create adversarial samples in adversarial defense?

[1] Zhang et al., Adversarial AutoAugment. ICLR, 2020.

---

> ### Author Response · Authors · 2022-08-02
> **Response to Reviewer JwsS: Part I**
>
> We appreciate your time and suggestions!  We have added new experiments on ImageNet and comparisons to other data augmentations in the new version of our paper. Here are our detailed replies to your questions.
>
> **(1) The paper can compare LP-A3 with other adversarial data augmentation [1] strategies in terms of their idea and performance, so we will know which approach is better in which scenarios.**
>
> LP-A3 differs with previous adversarial augmentations in various ways including augmentation strategies, objectives, and applicable scenarios:
>
> - **Augmentation strategies:** Adversarial Autoaugment [1] learns a policy to produce a sequence of augmentation operations, which are still pre-defined by human domain knowledge or heuristics and restricted to limited options. In contrast, LP-A3 is a prior-free and instance-specific auto-augmentation technique that produces augmentations by solving an optimization problem with a principal objective.
> - **Objective:** Adversarial Autoaugment's objective is to maximize the classification error, which is contrary to our label-preserving constraint. Adversarial Autoaugment might change the class information for the augmented data if the learned strength for the augmentation operation is too strong.
> - **Applicable scenarios:** Adversarial Autoaugment [1] is designed for supervised learning tasks and the natural image domain, while LP-A3 can be applied to a broad range of representation learning tasks and domains as demonstrated in our experiments.
>
> We have added discussion in the related work section and empirical comparison with Adversarial Autoaugment in Sec.D in the Appendix.
>
> **(2) The results on the large-scale ImageNet dataset will make this paper stronger.**
>
> In order to validate the performance of LP-A3 on large-scale datasets, we compare FixMatch and "FixMatch+LP-A3" on ImageNet the semi-supervised learning. ResNet-50 is used as the backbone network. We randomly choose 10\% data and set them as labeled data, while the remaining 90% are unlabeled data. The results are reported in the table below, in which LP-A3 can improve FixMatch by a large margin, especially during the early stage (>6\% at 100,000 iterations).  These additional results have been added to Section D of the Appendix in the revised version.
>
> ImageNet|100k Iter|250k Iter|400k Iter
> ---|:--:|---:|---:
> FixMatch|44.45|58.79|62.87
> FixMatch+LP-A3|51.22|60.15|63.67
>
> **(3) It would be better to compare LP-A3 with other Automated Data Augmentation methods, like AutoAugment, Adversarial AutoAugment, and sample-mixing augmentations, like Mixup, Cutmix, on the full dataset of popular classification benchmarks, e.g. CIFAR10 and CIFAR100.**
>
> - A great number of methods are mainly designed and tuned for the fully supervised learning on CIFAR10 and CIFAR100, leading to overfitting and poor true generalization performance as discussed in [2], especially for the state-of-the-art methods on these datasets. In this paper, we instead focus on those tasks deficient in or heavily relying on high-quality data augmentations, e.g., semi-supervised/noisy-label learning which lacks labeled or correctly labeled data, medical images in MedMNIST that lack sufficient domain knowledge.
> - Our method can be easily implemented for these three diverse and important categories of learning tasks and consistently improve all their performance --- This is a non-trivial contribution because in previous work every task needs to tune its own augmentation operations to reach the best performance, e.g., SimCLR and FixMatch. There are so many possible tasks in the world and we cannot expect that one single augmentation technique can achieve the best performance on all of them.
> - LPA3 does not aim at replacing existing hand-crafted data augmentation but is complementary to them.
> - We select Mixup and Adversarial AutoAugment as the representative of sample-mixing based augmentation and automated data augmentation respectively, and we empirically compare LP-A3 with them by applying LP-A3 on top of them. The results are reported in the table below, which shows that Mixup and Adversarial Autoaugment achieves high accuracy (>96\%) on CIFAR10 as they are designed for this task, but LP-A3 can further improve them, which indicates that LP-A3 are complementary to these previous data augmentations designed using domain knowledge. Note in the early stage the advantage of LP-A3 over Adversarial Autoaugment is especially significant (>0.7\%). These additional results have been added to Section D of the Appendix in the lately revised version.
>
> CIFAR10|50 Epochs|100 Epochs|150 Epochs|
> ---|:--:|---:|---:
> Mixup|87.39|93.74|96.05
> Mixup+LPA3|86.82|94.73|96.12
> Adversarial Autoaugment|89.70|95.05|97.09
> Adversarial Autoaugment+LPA3|90.45|95.33|97.20

---

> > ### Author Response · Authors · 2022-08-02
> > **Response to Reviewer JwsS: Part II**
> >
> > **(4) Prior works such as RandAugment run multiple times for the model and present the mean and standard deviation. Did the authors run multiple times for each model and what about presenting the mean and standard deviation?**
> >
> > We have already provided error bars (mean and std over three random trials) of semi-supervised learning in Table 1 of the paper, and we will provide those for noisy-label learning and MedMNIST in the next version.
> >
> > The table below reports the error bars on CIFAR10, which show small variances and significant improvements brought by our method instead of some randomness.
> >
> > CIFAR10|40|250|4000
> > ---|:--:|---:|---:
> > FixMatch|89.51$\pm$3.14|93.81$\pm$0.29|94.66$\pm$0.13
> > FixMatch+LPA3|92.39$\pm$1.21|94.03$\pm$0.31|95.11$\pm$0.17
> >
> > **(5) With the opposing objectives, can LP-A3 be integrated with adversarial training? Does LP-A3 affect the use of perturbation-based adversarial attacks to create adversarial samples in adversarial defense?**
> >
> > - LP-A3 differs from adversarial attacks (used in adversarial training) in several fundamental aspects, e.g., motivations, formulations, applications, etc., though their optimizations look similar and are both related to the adversarial idea. A primary difference is that LP-A3 enforces a label-preservation constraint while adversarial attacks aim at distorting the predicted labels. Hence, unlike adversarial attacks, LP-A3 augmentations do not cross the classification boundaries. These different properties are motivated by their targeted applications: LP-A3 targets at improving the generalization performance of representation learning, while adversarial training aims at improving adversarial robustness.
> > - Adversarial training with LP-A3 can be an interesting and promising future direction. LP-A3 can potentially result in a better trade-off between adversarial robustness and clean data accuracy since LP-A3 can generate label-preserved adversarial examples and we can control such trade-off by controlling the label-preserving strength in LP-A3.
> >
> > **(6) It seems that LP-A3 relies on the representation learning network to compute the LPIPS distance and softmax class probability in Eq. (4). During the early stages of training, when is not well trained, is it possible that it may generate incorrect training signals for computing the LPIPS distance and softmax class probability for the label-preserving constrain? How does the method deal with this problem especially for small and more complicated datasets?**
> >
> > - Our method does NOT rely on any pre-trained or auxiliary model to produce the data augmentations: **this is an advantage instead of a drawback for practical usage**, which does not always have the access to pre-trained models.
> > - The neural net classifier in earlier stages might not provide the optimal surrogate model but training with LP-A3 will **not make the classifier worse** since LP-A3 preserves the current classifier's prediction, although the classifier might be inaccurate.
> > - At the earlier stages in Figure (4), where the classifier could be inaccurate, LP-A3 still brings significant improvements comparable or even greater than the later-stage improvement.
> > - In Figure (4), as the training goes on, the classifier prediction **becomes more and more consistent** with the oracle label, thus providing a better approximation of the label-preservation constraint used to produce the data augmentations.
> >
> > ---
> > References:
> >
> > [1] Xinyu Zhang, Qiang Wang, Jian Zhang, Zhao Zhong. Adversarial AutoAugment. ICLR 2020.
> >
> > [2] Benjamin Recht, Rebecca Roelofs, Ludwig Schmidt and Vaishaal Shankar. Do CIFAR-10 Classifiers Generalize to CIFAR-10?

---

> ### Author Response · Authors · 2022-08-07
> **New experiments on ImageNet and comparisons to TWO new augmentations. Would you mind checking them and reconsidering the rating?**
>
> We are posting this message as a reminder and would appreciate any further discussions in public.
>
> We appreciate your constructive comments on our paper. We hope we have the opportunity to see your further comments, answer any additional questions, and ultimately improve the quality of our submission. We have made modifications to the paper. In particular,
>
> - As to your concern about the performance on large-scale datasets, we have **added new experiments on ImageNet** semi-supervised learning. Results show that LP-A3 still brings significant improvements to FixMatch on ImageNet, especially in the early stage.
> - As to your concerns on the comparison with other Augmentation methods on the full dataset of popular classification benchmarks, we have **added new experiments comparing LP-A3 with Adversarial Autoaugment and Mixup on CIFAR10**, the results show that LP-A3 can still bring further improvement on top of these state-of-the-art augmentation methods on CIFAR10, which indicates that LP-A3 are **complementary to these previous data augmentations designed using domain knowledge**.
> - As to your concern about error bars, we have **added error bars on semi-supervised learning** in the revised version and will add those for noisy-label learning and medical image classification in the next revision.
> - As to your concern about possible incorrect training signals for producing augmentations, we have revealed that even though the classifier might be inaccurate in the early stage, LP-A3 will not make the classifier worse but can instead bring improvement. Our method **does NOT rely on any pre-trained or auxiliary model** to produce the data augmentations: this is an **advantage for practical usage**,
>
> We appreciate the opportunity to further incorporate your suggestions to improve our work if you get a chance to read our response.

---

> > ### Comment · Reviewer_JwsS · 2022-08-08
> > **After Rebuttal**
> >
> > Thanks the authors for giving the explanation of baseline comparison and additional results. I appreciate the effort that the authors put into addressing my concerns. I believe that the above results and discussion can improve the quality of the work; hence I raise my score to 5 from 4.
> > However, I still think there is some room for improvement. For example:
> > - In response (5), the authors explain the difference between LP-A3 and adversarial training. While the two methods have different targets, generalization performance and adversarial robustness are both important considerations in practice. As both LP-A3 and adversarial training use image perturbation to create adversarial samples or augmented samples, I still believe that using LP-A3 may preclude common adversarial training methods. Therefore, I recommend that the authors mention it as a limitation in the manuscript.
> > - In response (6), the authors argue that the inaccurate LPIPS and predicted probabilities would not worsen the classifier in four experiments (Fig. 4). The claim would be stronger if supported by more evidence. For example, using different pre-trained models to compute the LPIPS and class probabilities and study the effects of the end classifier.

---

> > > ### Author Response · Authors · 2022-08-09
> > > **Thank you for your suggestion!**
> > >
> > > We appreciate the reviewer for taking the time to review our responses and make valuable suggestions. We will discuss more about the relationship to adversarial training and keep running more experiments using different pre-trained models to improve the quality of the work.

---

### Official Review · Reviewer_sdEW · 2022-07-08

**Rating:** 4
**Confidence:** 4
**Soundness:** 3 good
**Presentation:** 3 good
**Contribution:** 3 good

**Summary:**

The paper proposes a data augmentation strategy without any pre-defined augmentation operations. The goal of the data augmentation is to create a distant hard positive example while preserving the label. The paper uses a surrogate, based on LPIPS distance and model predictions, to find the optimal augmentation for an image. The experiments are conducted on supervised, semi-supervised, and noisy-label learning.

**Questions:**

The main questions are presented in Strengths And Weaknesses, but there are some more questions that I am curious about.
1. For Medical Image Classification, AutoAugment might be too strong. Some common data augmentations might have good results, such as random crop+colorjitter+random horizontal flipping, used in Mean Teacher.

**Ethics Review Area:**

["I don’t know"]

**Limitations:**

Not discussed.

**Strengths And Weaknesses:**

strengths:
1. The auto-augmentation idea is clearly presented and easy to follow.
2. The paper provides a strong theoretical derive to the formula of LP-A3 (Eq.2), where the augmentation should contain less information about the original image and preserve the label.
3. The LP-A3 method achieves significant improvement on three different learning tasks.

weakness:
1. The implementation of the optimization problem has some obvious drawbacks. The paper uses the neural net classifier to approximate the conditional entropy of Y given X. However, the prediction of the classifier can be incorrect. Therefore, there might be some augmentation X' has the same classifier prediction as X, while the oracle label of X' is different from X, then the augmentation X' is undesired. On the other hand, there exists some augmentation X' has the same oracle label as X, while classifier prediction of X' is very different from X, but these augmentations are not considered by the introduced implementation. Virtual Adversarial Training considers the second situation and shows that augmentation should change the classifier prediction. So making the augmentation preserving the classifier prediction might not be the optimal choice.
2. The experimental result of semi-supervised learning is not very convincing. The reported performance of FixMatch is lower than the results in the original paper of FixMatch. The result can have some variance, especially in semi-supervised learning with few labels. It is possible the reported result is selected from multiple random seeds. The author should explain the unsatisfactory reproduction and report the variance of results for multiple random seeds.
3. It is not clear that the proposed LP-A3 itself can improve Semi-supervised Learning and Noisy-label learning. FixMatch uses RandAugment and CTAugment for data augmentation. The paper should test to replace all data augmentation in FixMatch with the proposed LP-A3.

---

> ### Author Response · Authors · 2022-08-02
> **Response to Reviewer sdEW:**
>
> We appreciate your time and suggestions! Here are our detailed replies to your questions.
>
> **(1) The implementation of the optimization problem has some obvious drawbacks ...the prediction of the classifier can be incorrect.**
>
> - Our method does NOT rely on any pre-trained or auxiliary model to produce the data augmentations: **this is an advantage instead of a drawback for practical usage**, which does not always have the access to pre-trained models.
> - The neural net classifier in earlier stages might not provide the optimal surrogate model but training with LP-A3 will **not make the classifier worse** since LP-A3 preserves the current classifier's prediction, although the classifier might be inaccurate.
> - At the earlier stages in Figure (4), where the classifier could be inaccurate, LP-A3 still brings significant improvements comparable or even greater than the later-stage improvement.
> - In Figure (4), as the training goes on, the classifier prediction **becomes more and more consistent** with the oracle label, thus providing a better approximation of the label-preservation constraint used to produce the data augmentations.
> - On the contrary, it is more likely for Virtual Adversarial Training (VAT) to change the oracle label of $X^\prime$ as it changes the classifier prediction of $X^\prime$, and the classifier prediction is consistent with the oracle label in the late stage of training.
>
> **(2) The experimental result of semi-supervised learning is not very convincing.**
>
> The reason why our reproduction of FixMatch is slightly lower than the original paper is that the original paper of FixMatch has to train the model for 1,048,576 iterations to reach its reported accuracy, which is **too time-consuming**. In contrast, as shown in Table 1, applying LP-A3 augmentations can reduce the number of iterations from 1,048,576 to 262,144 to reach a comparable accuracy. A key advantage of LP-A3 for semi-supervised learning is to **speed up its slow convergence** especially during the early stage, as shown in Figure 4(a).
>
> The table below reports the error bars on CIFAR10, where we can see that the variance is small and our improvement is significant, which does not come from randomness.
>
> CIFAR10|40 |250 |4000
> ---|:--:|---:|---:
> FixMatch|89.51$\pm$3.14|93.81$\pm$0.29|94.66$\pm$0.13
> FixMatch+LPA3|92.39$\pm$1.21|94.03$\pm$0.31|95.11$\pm$0.17
>
> We have already provided error bars (mean and std over three random trials) of semi-supervised learning in Table 1 of the paper, and we will provide those for all the noisy-label learning and MedMNIST experiments in the next version.
>
> **(3) The paper should test to replace all data augmentation in FixMatch with the proposed LP-A3.**
>
> LP-A3 does not aim at replacing existing hand-crafted data augmentations because they can provide complementary information to the training, i.e., LP-A3 produces model-adaptive augmentations without using any human expert priors while the hand-crafted data augmentations provide human priors:
>
> - LP-A3 and existing hand-crafted data augmentations are two different types of data augmentations that encode complementary knowledge for training: most existing augmentations encode human experts' domain knowledge (e.g., invariance of the model output under pre-defined transforms like horizontal flips for natural images) and they are not instance-specific (i.e., they apply the same transforms to all data). In contrast, LP-A3 focuses on the knowledge and the task-feedback from the model itself during training and uses them to optimize a model-adaptive and instance-specific augmentation. LP-A3 does not encode any domain knowledge or human priors.
> - As demonstrated in our experiments, although RandAugment and CTAugment are selected and/or designed specifically for semi-supervised learning and they already bring significant improvements, LP-A3 can still further improve the performance on top of them, which is a noteworthy contribution and indicates that LP-A3 is not redundant but complementary to those existing augmentations.
> - In Table 3 (the experiments on medical image classification tasks from MedMNIST), we have evaluated LP-A3's performance when no other data augmentation or human expert is available. LP-A3 itself can improve performance in such challenging but practical scenarios even without any human expert knowledge.
>
> **(4) For Medical Image Classification,... Some common data augmentations might have good results.**
>
> We evaluate your suggested common augmentations (random crop+colorjitter+random horizontal flipping) on MedMNIST and find that they achieve promising performance. **However, LP-A3 can bring non-trivial improvements on top of them**, according to the results reported in the table below.
>
> Augmentation|DermaMNIST|BloodMNIST|OrganCMNIST|OrganSMNIST|
> ---|:--:|---:|---:|---:
> Common|76.14|96.81|91.78|81.58
> Common+LPA3|76.22|96.97|92.42|81.87

---

> ### Author Response · Authors · 2022-08-07
> **Can you provide the reasons for downgrading the score? We have addressed all you concerns! LP-A3 does NOT rely on any auxiliary model: this is an advantage for practical usage.**
>
> Would you mind elaborating on your reasons for downgrading the score after we addressed all your concerns? We have addressed all your concerns with detailed responses and new experiments!
>
> - As to your concerns about using neural net classifier to provide approximation, we have revealed that even though the classifier might be inaccurate in the early stage, LP-A3 will not make the classifier worse **by preserving the current classifier's prediction** but can instead bring improvement, as shown in Figure 4(a). Our method does NOT rely on any pre-trained or auxiliary model to produce the data augmentations: this is an advantage for practical usage,
>
> - The reason why our reproduction of FixMatch is slightly lower than the original paper is that we train all the models for only 262,144 iterations, in order to highlight that LP-A3 can **speed up the slow convergence of semi-supervised learning**, especially during the early stage.
>
> - As to your concern about error bars, we have **added error bars on semi-supervised learning** in the revised version and will add those for noisy-label learning and medical image classification in the next version. We did NOT select results from multiple random seeds.
>
> - As to your concern about the performance of LP-A3 alone, LP-A3 **does NOT aim at replacing** existing hand-crafted data augmentations but are complementary to them, i.e., LP-A3 produces model-adaptive augmentations without using any human expert priors. Moreover, RandAugment and CTAugment used by FixMatch are selected and/or designed specifically for semi-supervised learning and they already bring significant improvements, **there is NO reason not using them**! In Table 3 (medical image classification tasks), we **have evaluated LP-A3's performance when no other data augmentation** or human expert is available.
>
> We appreciate your time and comments! However, we respectfully disagree with your decision of downgrading without discussing the new concerns leading to the downgrade. Would you mind adding more details about the reasons behind it? Thanks！

---

### Official Review · Reviewer_vSqH · 2022-07-12

**Rating:** 8
**Confidence:** 3
**Soundness:** 4 excellent
**Presentation:** 3 good
**Contribution:** 4 excellent

**Summary:**

The paper proposes a new prior free data augmentation strategy while usual approaches rely on prior knowledge of elementary transformations that preserve the labeling. The method is cast as an optimization problem that aims at producing a hard positive example for every training sample. Augmentation data are refined epoch after epoch as the learning of the model progresses.

The method does not require learning an additional generative model and may be used in various tasks such as supervised learning, semi supervised learning and noisy label learning.
The method is well funded. It is derived from a formalization of the process of data augmentation as a probabilistic graphical model and of the formalization of what should be a good representation and what should be a good augmentation strategy. Details are provided on how to implement the strategy and experimental results are reported on few datasets and for different tasks, n supervised learning semi supervised learning, noisy label learning.

**Questions:**

Section 3 line 132 it is said « We decouple the randomness into two parts… ». But then there is only one random variable N? and X is a deterministic function of N and of Y. What is the second part of randomness ?

Section 4 build on the notion of sufficient representation but i don’t see a clear definition of this.

Section 5.2. Maybe you could add some details about TCS, which you only mention, but which is part of your method (cf. Algorithm 1).

Datasets. What are the sizes of the datasets used in the experiments ?

**Strengths And Weaknesses:**

The approach looks innovative with a refinement of the augmentation data from which the model is learned, all along the learning process. I am not a specialist of data augmentation but it does not seem to be the mainstream approach. On the opposite badine methods usually rely on prior knowledge of label preserving transformations, and most often are not learned using a task feedback.

The paper is well written and the approach is well detailed put to implementation details.

Experimental  results show significant and consistent improvement of the methods over the baselines that look well chosen for the different tasks that are investigated.

Some parts could be more detailed. In particular section 4 builds on the notion of sufficient representation but i don’t see a clear definition of this and i did not fully get the definition 4.0.1.  In the algortihm 1, i am not sure if selecting samples for which augmentation will be considered is a way of accelerating the learning or is necessary ti get good results. In the experimental section main ideas underlying baselines could have been explained (e.g. FixMatch [35])  at least the main common features and main differences with the proposed method could be discussed.

---

> ### Author Response · Authors · 2022-08-02
> **Response to Reviewer vSqH:**
>
> We thank the reviewer for their positive and insightful review. Here are our detailed replies to your questions.
>
> **(1) Section 3 line 132 it is said « We decouple the randomness into two parts… ». But then there is only one random variable N? and X is a deterministic function of N and of Y. What is the second part of randomness ?**
>
> The second part of randomness comes from the random variable $Y$ for the label, which takes values in the discrete set of labels. An example $X$ is generated from two random variables: the label variable $Y$, and the nuisance variable $N$. To determine an example $X$, we need a realization of $Y$ (e.g., "cat") and a realization of $N$ (e.g. background, color, etc). This decomposition follows the theory in Alessandro et al. [1].
>
>
> **(2) Section 4 build on the notion of sufficient representation but I don’t see a clear definition of this.**
>
> Thank you for pointing out this! We have updated Definition 4.0.1 in the revised version to include a rigorous definition of sufficient representation before defining the $\epsilon$-minimal sufficient representation. In short, $Z$ is sufficient if $I(Z \wedge Y) = I(X \wedge Y)$, i.e., $Z$ keeps the information about $Y$.
>
> **(3) Section 5.2. Maybe you could add some details about TCS, which you only mention, but which is part of your method (cf. Algorithm 1).**
>
> TCS[2] measures the consistency of the model output (i.e., the predicted class distribution) for a sample over historical training steps. We select top $\tau$% sample with the lowest TCS to apply our data augmentation because samples with small TCS tend to have sharp loss landscapes. These samples provide more informative gradients than others and applying our model-adaptive data augmentations can bring more improvement to their representation invariance and loss smoothness. In Figure (5), we conduct a thorough sensitivity analysis on $\tau$ over three tasks and find that sample selection with TCS can effectively improve the performance. Moreover, in this way, we do not need to apply our augmentation to every training sample and thus save the training cost. We have added details about TCS in Sec.B of the Appendix.
>
> **(4) Datasets. What are the sizes of the datasets used in the experiments?**
>
> The CIFAR-10 (CIFAR-100) dataset contains 60,000 images of size 32x32 from 10 (100) classes.The STL-10 dataset contains 5,000 labeled images of size 96×96 from 10 classes and 100,000 unlabeled images. The datasets in MedMNIST contain from 10,000 to 236,386 images of size 28x28. During rebuttal, we added experiments on ImageNet which contains 1,281,167 training images and 50,000 validation images of size 256x256.
>
> ---
> References:
>
> [1] Alessandro Achille and Stefano Soatto. Emergence of invariance and disentanglement in deep representations. The Journal of Machine Learning Research. 2018.
>
> [2] Tianyi Zhou, Shengjie Wang, and Jeff A. Bilmes. Time-Consistent Self-Supervision for Semi-Supervised Learning. International Conference on Machine Learning (ICML), 2020.

---

### Meta-Review · Area_Chair_wqxd · 2022-08-24

**Recommendation:** Accept
**Confidence:** Certain

**Metareview:**

This paper presents a new augmentation method (though it's similar to adversarial training) that significantly improves various benchmarks, methods, and tasks. Besides, the authors provide an information-theoretic ground for the proposed method. AC appreciate the technical contribution to the community. After rebuttal, the authors addressed most of the concerns. AC recommends accept.

AC also would like to suggest the authors comprehensively compare your method with adversarial training in revision.

**Award:**

No

---

### Decision · Program_Chairs · 2022-09-14

Accept